# The subiculum is a patchwork of discrete subregions

**Mark S Cembrowski\*, Lihua Wang, Andrew L Lemire, Monique Copeland, Salvatore F DiLisio, Jody Clements, Nelson Spruston**

Janelia Research Campus, Howard Hughes Medical Institute, Ashburn, United States

**Abstract** In the hippocampus, the classical pyramidal cell type of the subiculum acts as a primary output, conveying hippocampal signals to a diverse suite of downstream regions. Accumulating evidence suggests that the subiculum pyramidal cell population may actually be comprised of discrete subclasses. Here, we investigated the extent and organizational principles governing pyramidal cell heterogeneity throughout the mouse subiculum. Using single-cell RNA-seq, we find that the subiculum pyramidal cell population can be deconstructed into eight separable subclasses. These subclasses were mapped onto abutting spatial domains, ultimately producing a complex laminar and columnar organization with heterogeneity across classical dorsal-ventral, proximal-distal, and superficial-deep axes. We further show that these transcriptomically defined subclasses correspond to differential protein products and can be associated with specific projection targets. This work deconstructs the complex landscape of subiculum pyramidal cells into spatially segregated subclasses that may be observed, controlled, and interpreted in future experiments.

DOI: https://doi.org/10.7554/eLife.37701.001

## Introduction

To interpret the complexity of the brain, neuroscience has sought to deconstruct brain regions and circuits into elemental and interpretable cell types (*Zeng and Sanes, 2017*). Historically, this deconstruction has employed morphological and electrophysiological approaches, giving rise to the classical cell-type definitions that broadly delineate cells in the brain. Modern neuroscientific tools now enable high-throughput interrogation of complementary modalities, including gene expression and connectivity, to further partition and refine these cell types. Ultimately, a unified deconstruction of the nervous system will require projecting such modern, neurobiologically relevant elaborations onto classical cell types.

The hippocampus of the mammalian brain provides a comprehensively studied brain region to identify such cell-type-specific elaborations and relate them to function. This brain region has been studied extensively for its critical roles in episodic memory (*Scoville and Milner, 1957*), spatial navigation (*O'Keefe and Nadel, 1978*), and emotionally motivated behavior (*Kjelstrup et al., 2002*). To date, evidence is emerging that suggests heterogeneity within classical cell types of the hippocampus may be an important feature for mediating hippocampal computation and function (*Cembrowski et al., 2016a*; *Cembrowski et al., 2016b*; *Danielson et al., 2016*; *Igarashi et al., 2014*; *Knierim et al., 2014*; *Lee et al., 2015*; *Lee et al., 2014*; *Soltesz and Losonczy, 2018*; *Strange et al., 2014*; *Thompson et al., 2008*).

One of these classical cell types is the pyramidal cell type of the subiculum, which acts as an output from the hippocampus to a wide array of downstream targets (*Aggleton and Christiansen, 2015*; *Naber and Witter, 1998*). We recently found that the dorsal pole of the subiculum can be partitioned into distinct proximal and distal subregions (*Cembrowski et al., 2018*), motivating us to

**\*For correspondence:** cembrowskim@janelia.hhmi.org

**Competing interests:** The authors declare that no competing interests exist.

investigate whether additional heterogeneity could be revealed when considering the full spatial extent of the subiculum. Indeed, recent investigations using immunohistochemical labeling argue that the proximal subiculum is composed of a molecular layer and multiple cell body layers, each distinguished by molecular and morphological differences, while the distal subiculum is more uniform (*Ishihara and Fukuda, 2016*). Additionally, as specific downstream projections and postulated functional contributions change across space in the subiculum (*Böhm et al., 2018*; *Bubb et al., 2017*; *Ishihara and Fukuda, 2016*; *Naber and Witter, 1998*; *O'Mara et al., 2009*), understanding subicular organizational rules will likely be critical for a cell-type-specific deconstruction of memory, cognition, and emotion.

Here, we took a multimodal approach to understanding the organizational logic of the subiculum. Using single-cell next-generation RNA sequencing (scRNA-seq), we found that subiculum pyramidal cells could be partitioned into eight subclasses. We were able to register these subclasses in space, uncovering a patchwork landscape of subicular subfields. We subsequently mapped these subfields onto specific protein products and projection targets. We provide these scRNA-seq data, in conjunction with analysis and visualization tools, as a public resource. In total, this work produces a multimodal deconstruction of a key brain region, and will serve as a foundation for continuing to unravel the cell-type-specific rules of cognition.

## Results

### Overview of subiculum scRNA-seq atlas: construction, validation, and extension

We took two complementary approaches to obtain cells for our subiculum scRNA-seq atlas (overview: *Figure 1*; initial analysis: *Figure 2*). In one set of experiments, we microdissected out dorsal, intermediate, and ventral regions of the subiculum from wild-type mice (n = 3 mice total, one mouse per region). We dissociated these subiculum regions and manually selected cells for sequencing. In a second set of experiments, we injected retrograde beads into subiculum targets, labeling specific projection classes of subiculum cells (n = 3 mice total, one mouse per projection class). In these experiments, the subiculum was microdissected and dissociated, and manual selection was used to specifically purify for labeled cells. In both experiments, library preparation, sequencing, and analysis were handled according to previous methods (*Cembrowski et al., 2018*) (see Materials and methods).

This approach obtained high-read-depth, high-quality transcriptomes from 1150 cells (5.6 ± 1.0 thousand expressed genes/cell, mean ± SD). Data from these cells, in conjunction with user-friendly analysis and visualization tools, are available on http://hipposeq.janelia.org. To ensure that the results and conclusions of our scRNA-seq analysis were robust and predicted higher order features, we validated predictions from this dataset with additional biological replicates (*Figure 2—figure supplement 3*) and cross-validated and extended our findings using *in situ* hybridization (*Figures 3–7*), immunohistochemistry (*Figure 8*), and projection mapping (*Figure 9*).

### The transcriptomic landscape of the subiculum

To begin, we computationally pooled all of our transcriptomes, analyzing our datasets agnostic to selection method (i.e. unlabeled WT cells vs. labeled projection cells). We performed clustering using a graph-based clustering approach (*Satija et al., 2015*) (see Materials and methods), and visualized clusters through t-SNE-based nonlinear dimensionality reduction (*Figure 2A*; see also *Figure 2—figure supplement 1A* for principal component analysis). From this analysis, nine clusters were identified that expressed marker genes associated with excitatory neurons (e.g. *Camk2a*, *Slc17a7*; *Figure 2A,B*, *Figure 2—figure supplement 1*; note 14 putative non-neuronal cells and 13 putative interneurons were excluded from analysis, see Materials and methods). These clusters were robust, as using a supervised random forest classifier illustrated that 400 cells (~36% of dataset; 1103 total cells in dataset) were sufficient for ~80% success in predicting cluster identity (*Figure 2—figure supplement 1B*).

Remarkably, single genes were largely sufficient to delineate individual clusters (*Figure 2B*). Relatively large subclasses of cells were delineated by the marker genes *S100b*, *Dlk1*, *Tpbg*, and *Fn1*. Smaller subclasses, putatively corresponding to rarer subclasses of excitatory cells, showed

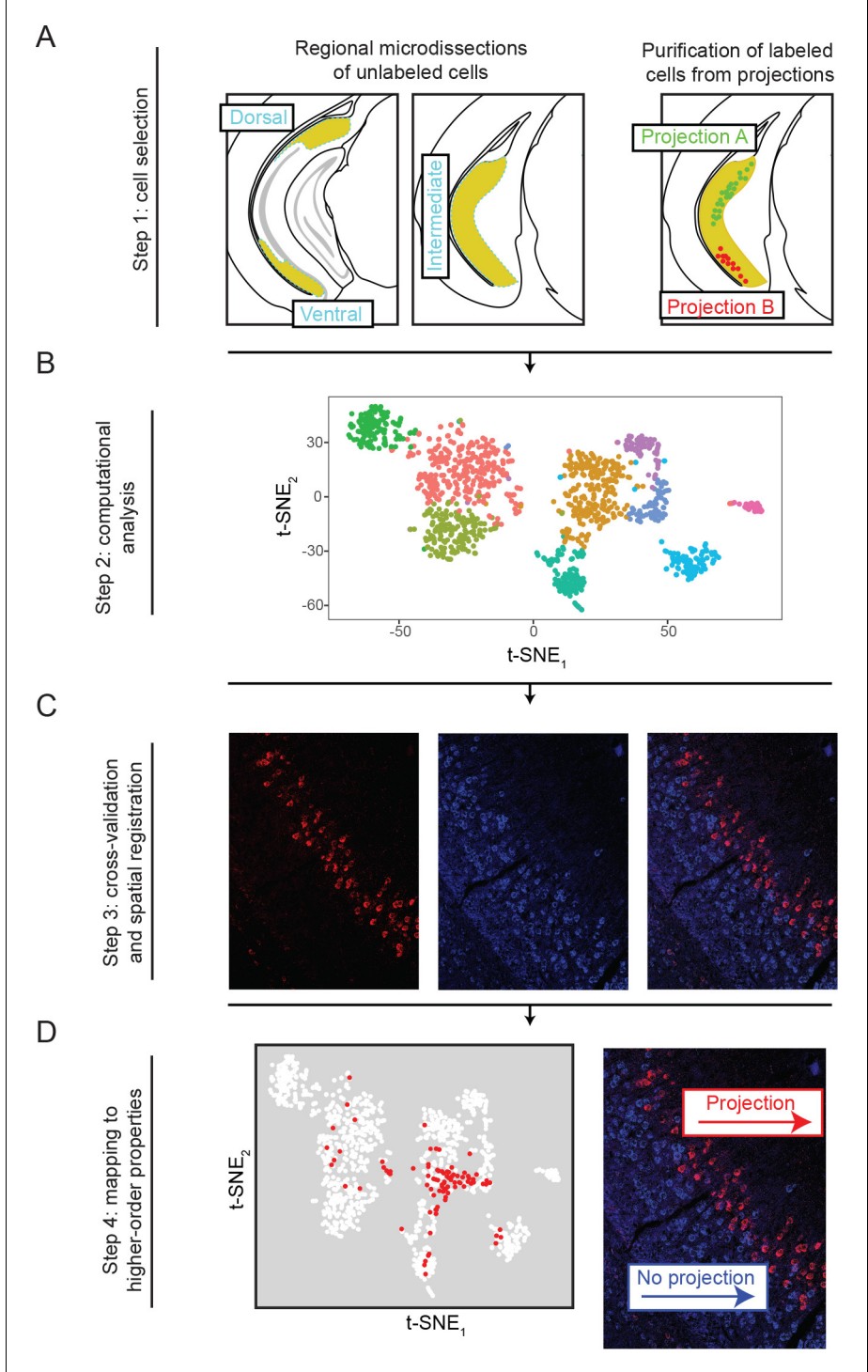

**Figure 1.** Overview of the generation, validation, and extension of the transcriptomic landscape of subiculum pyramidal cells. (**A**) Two strategies, based upon geography and projections, were used to select cells for scRNA-seq. (**B**) Single-cell transcriptomes were constructed and analyzed. (**C**) Subclasses revealed by scRNA-seq were cross-validated and spatially registered by *in situ* hybridization. (**D**) Higher order features (e.g. projection classes) were mapped onto subclasses.

DOI: https://doi.org/10.7554/eLife.37701.002

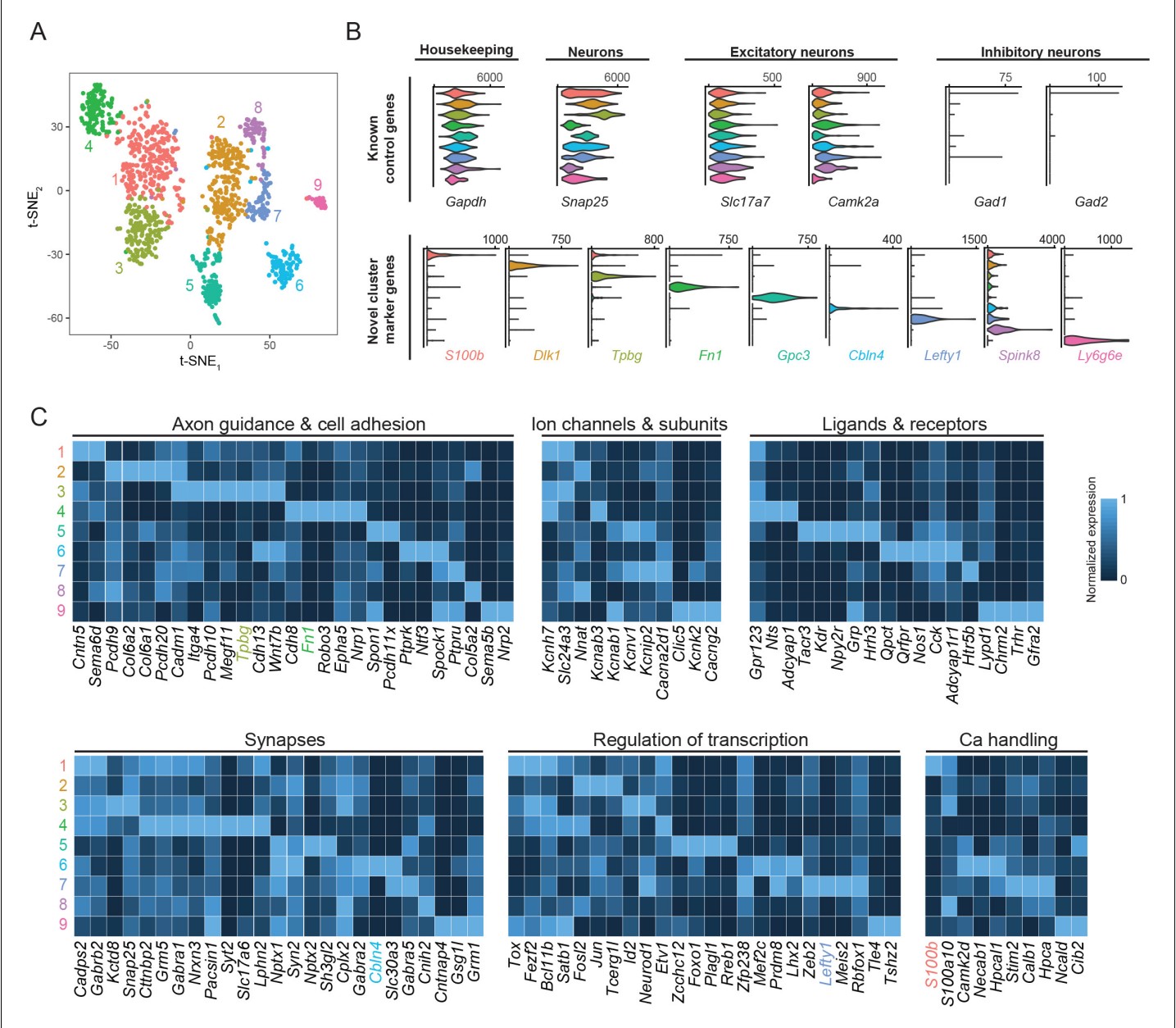

**Figure 2.** Subiculum pyramidal cells are divisible into transcriptomic subclasses. (**A**) Gene expression across cells of the subiculum, visualized by t-SNE. Colors indicate cluster identified by graph-based clustering, with cluster number provided alongside. (**B**) Expression of control genes and cluster-specific marker genes, summarized across clusters. Results are depicted as violin plots, which illustrate the smoothed distribution of expression across all cells. (**C**) Heatmap of genes with neuronally relevant ontologies that are enriched or depleted in individual clusters. Marker genes that correspond to specific ontologies are colored according to their respective cluster. Note that some marker genes (specifically *Dlk1*, *Gpc3*, *Spink8*, *Ly6g6e*) do not correspond to the ontologies shown here.

DOI: https://doi.org/10.7554/eLife.37701.003

The following figure supplements are available for figure 2:

**Figure supplement 1.** Expanded computational analysis of scRNA-seq data.
DOI: https://doi.org/10.7554/eLife.37701.004

**Figure supplement 2.** Hierarchical organization and differentially expressed genes.
DOI: https://doi.org/10.7554/eLife.37701.005

**Figure supplement 3.** Reproducibility of clusters across biological replicates.
DOI: https://doi.org/10.7554/eLife.37701.006

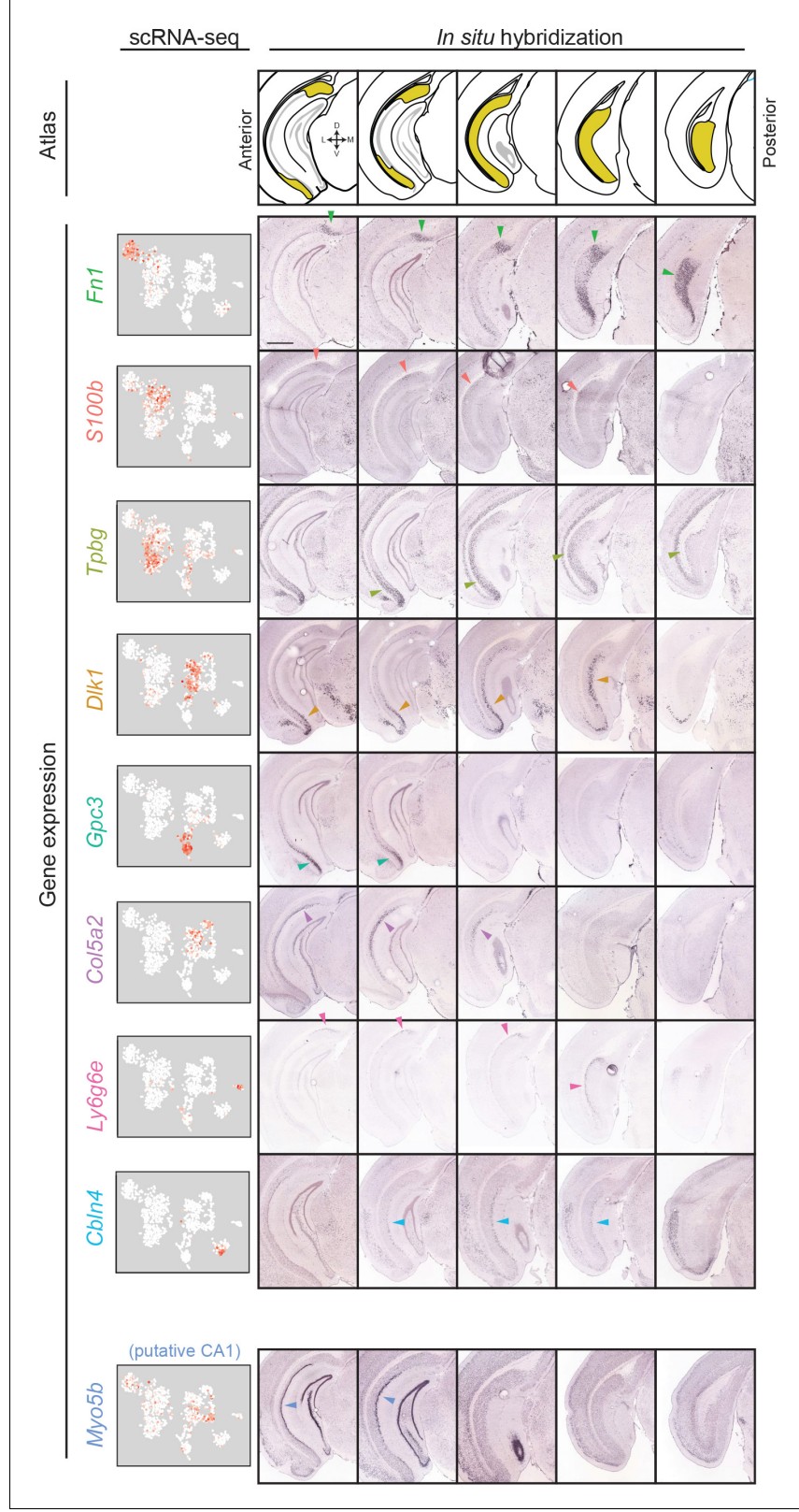

**Figure 3.** Gene expression clusters map onto distinct spatial domains in the subiculum. For each transcriptomic cluster, expression of a corresponding marker gene is shown across the anterior-posterior axis of the subiculum. Arrows indicate example regions of dense expression referred to in main text. Atlas images illustrate subiculum colored in yellow (atlas images, here and elsewhere, modified from *Paxinos and Franklin, 2004*), with cardinal

*Figure 3 continued on next page*

*Figure 3 continued*

directions corresponding to dorsal, ventral, medial, and lateral directions. scRNA-seq images illustrate expression colored from white to red on a logarithmic scale. Histological images illustrate coronal sections from the Allen Brain Atlas (*Lein et al., 2007*). Scale bar: 1 mm.
DOI: https://doi.org/10.7554/eLife.37701.007

The following figure supplements are available for figure 3:

**Figure supplement 1.** Proximal and distal subiculum, as viewed through coronal and horizontal sections.
DOI: https://doi.org/10.7554/eLife.37701.008
**Figure supplement 2.** Expression of *Myo5b* is associated with CA1 pyramidal cells.
DOI: https://doi.org/10.7554/eLife.37701.009
**Figure supplement 3.** No clusters are associated with genes expressed in parasubiculum, presubiculum, or postsubiculum.
DOI: https://doi.org/10.7554/eLife.37701.010
**Figure supplement 4.** Examination of gene expression associated with immunohistochemically identified subdomains.
DOI: https://doi.org/10.7554/eLife.37701.011

expression of *Gpc3*, *Cbln4*, *Lefty1*, *Spink8,* and *Ly6g6e*. In addition to these marker genes, a host of differentially expressed genes that spanned critical neuronal functions were also identified (e.g. axon guidance and cell adhesion, ion channels and associated subunits, ligands and receptors, regulation

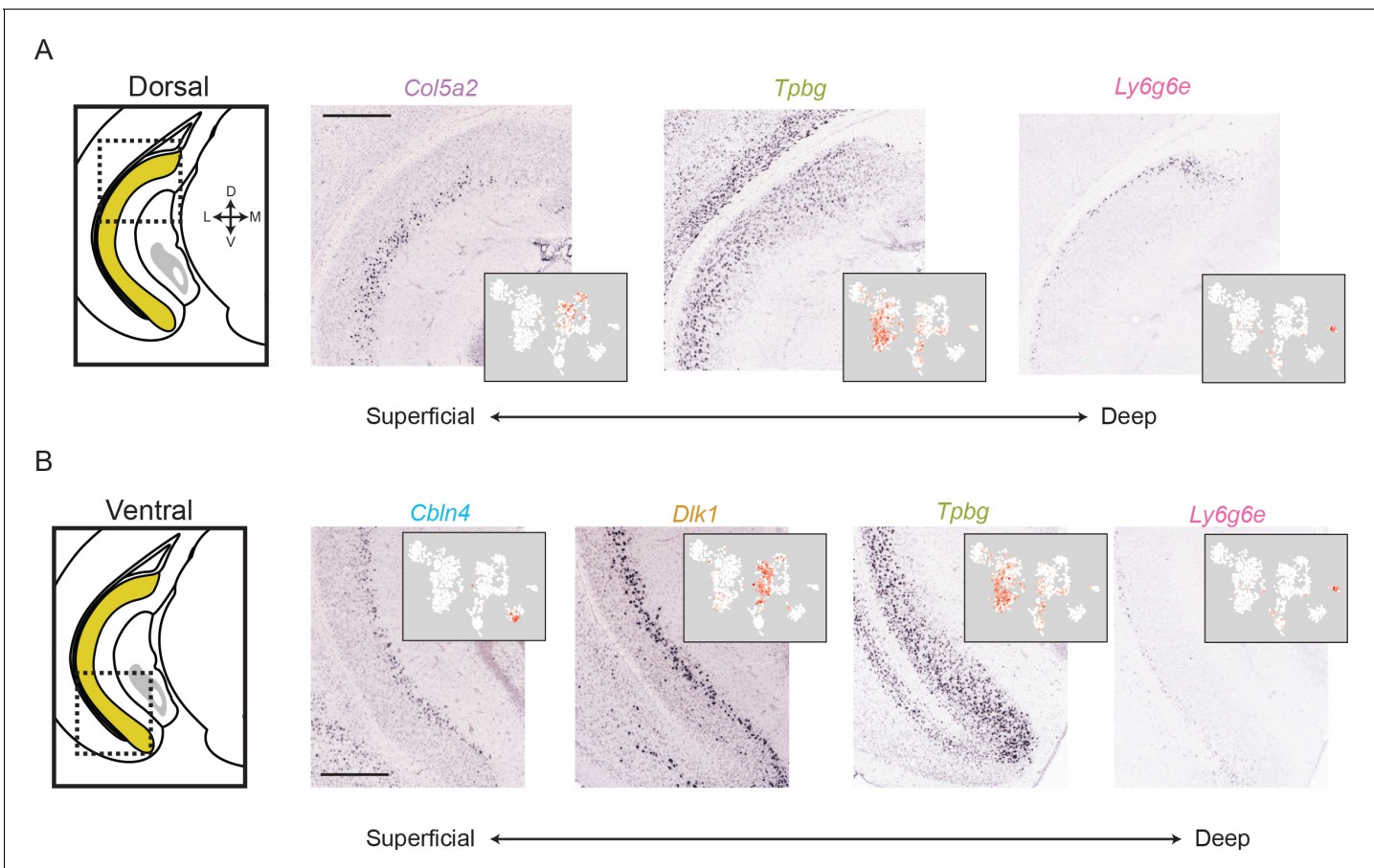

**Figure 4.** The subiculum can be deconstructed into distinct lamina across the long axis. (**A**) For a dorsal region of the subiculum (atlas at left), marker gene expression exhibits a superficial-to-deep lamination pattern. Scale bar: 500 μm. (**B**). As in A, but for marker gene expression in the ventral subiculum. Scale bar: 500 μm.
DOI: https://doi.org/10.7554/eLife.37701.012

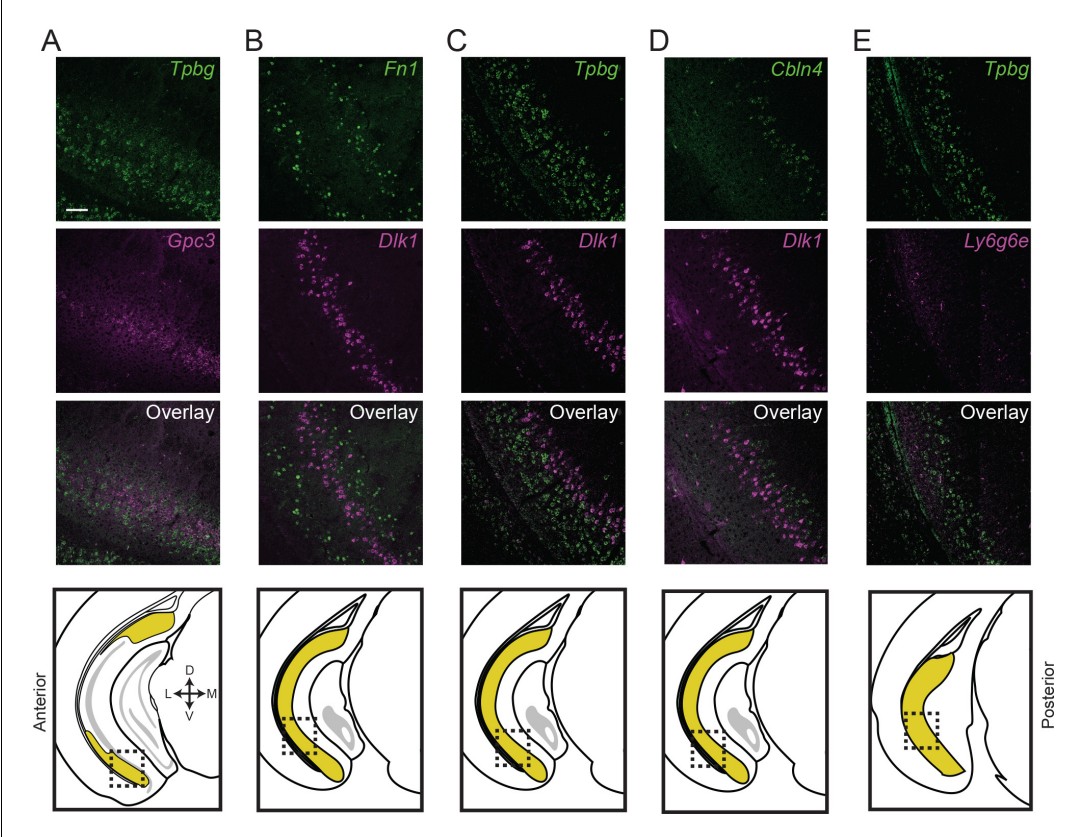

**Figure 5.** Subiculum subclasses exhibit discrete, abutting boundaries. (A) Two-color fluorescent ISH detecting expression of *Tpbg* and *Gpc3* marker genes, directly illustrating subiculum subdomains are abutting and non-overlapping. Atlas schematic in lowest row denotes area examined. (B-E) As in (A), but for *Fn1* and *Dlk1* (B), *Tpbg* and *Dlk1* (C), *Cbln4* and *Dlk1* (D), and *Tpbg* and *Ly6g6e* (E). Scale bars: 100 μm.
DOI: https://doi.org/10.7554/eLife.37701.013

of transcription, and calcium handling; *Figure 2C*). On average, a given cluster exhibited enrichment of 50 ± 32 genes relative to the remaining dataset (defined as >3 fold enriched on average and $p_{ADJ}$ <0.05; see Materials and methods), and 114 ± 68 genes when restricting analysis to pairwise cluster comparisons (*Figure 2—figure supplement 2*; *Supplementary file 2* and *3*). Notably, our analysis did not rely on any of these functional categories *a priori*, but rather recovered them from an unbiased approach. In total, these results illustrate that gene expression variation within subiculum excitatory cells is extensive, and likely underpins its functional heterogeneity.

## Replicate cross-validation

To examine the generalization of these results across biological replicates (i.e. animals), we next examined whether the same clusters were recapitulated across additional mice (n = 5 additional animals in total; see Materials and methods). From these animals, we dissected the subiculum and gathered data from a total of 847 excitatory neurons (5.6 ± 0.8 thousand genes expressed/cell; see Materials and methods). We performed analysis of this dataset identically to our previous dataset, and obtained eight clusters (*Figure 2—figure supplement 3A*). All eight clusters had marker genes associated with clusters obtained from our original dataset (geometric mean $p_{ADJ}$ values for cluster-specific markers = 8.7e-40, cf. $p_{ADJ}$ = 4.1e-62 from original dataset; *Figure 2—figure supplement 3A,B*). The single cluster from original dataset that was missed in the replicate dataset, associated with *Cbln4* expression, was detected in a subset of cells that separated in t-SNE space but failed to cluster at our predetermined resolution (*Figure 2—figure supplement 3C*). Importantly, no new clusters emerged from this replicate dataset, illustrating that our original scRNA-seq dataset

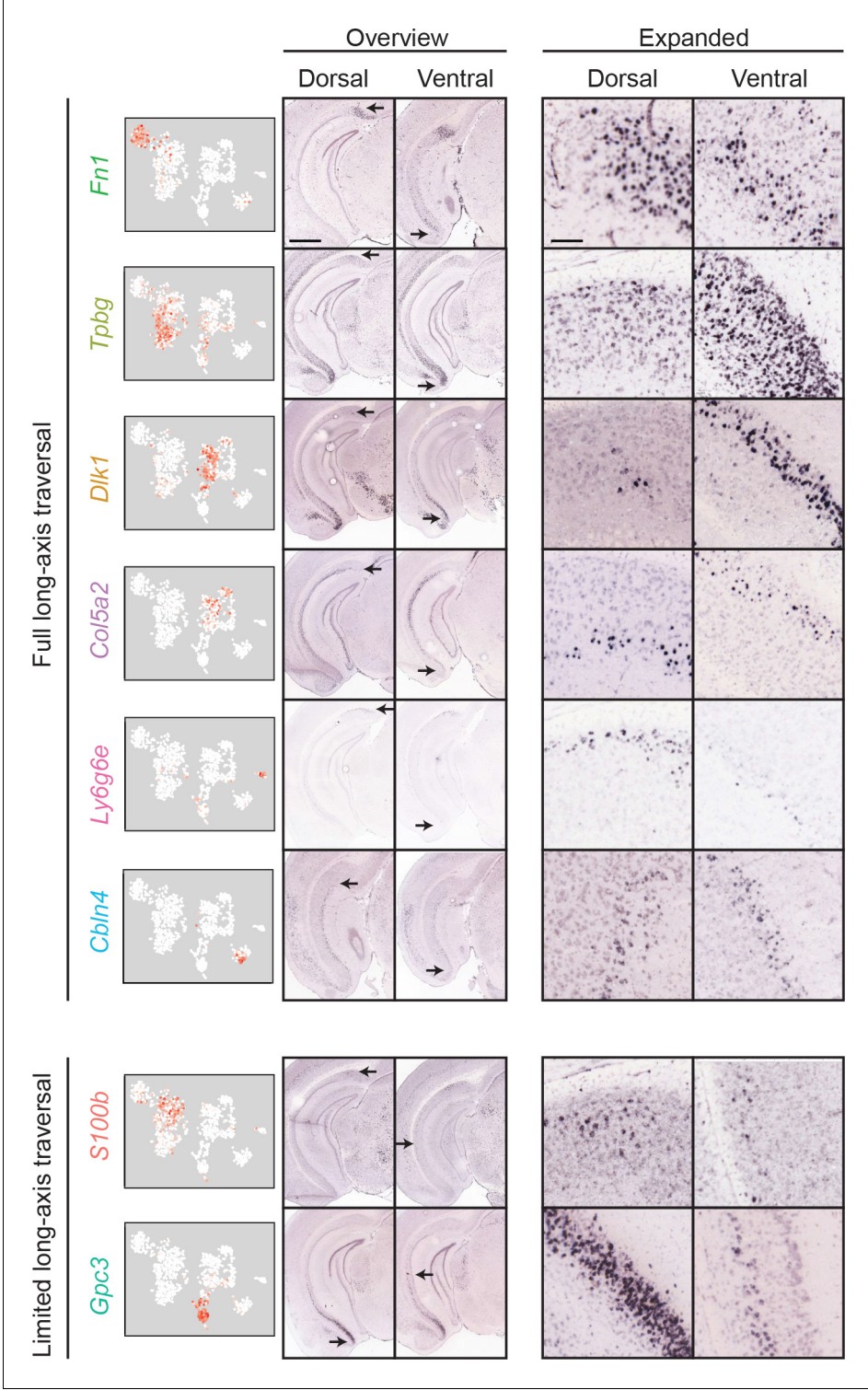

**Figure 6.** Most clusters span the full extent of the long axis. First column: scRNA-seq clusters. Second and third columns: for each cluster, the dorsal (second column) and ventral (third column) extent of marker gene expression are indicated. Scale bar: 1 mm. Fourth and fifth columns: expanded illustration of the areas denoted by arrows. Scale bar: 100 μm.

DOI: https://doi.org/10.7554/eLife.37701.014

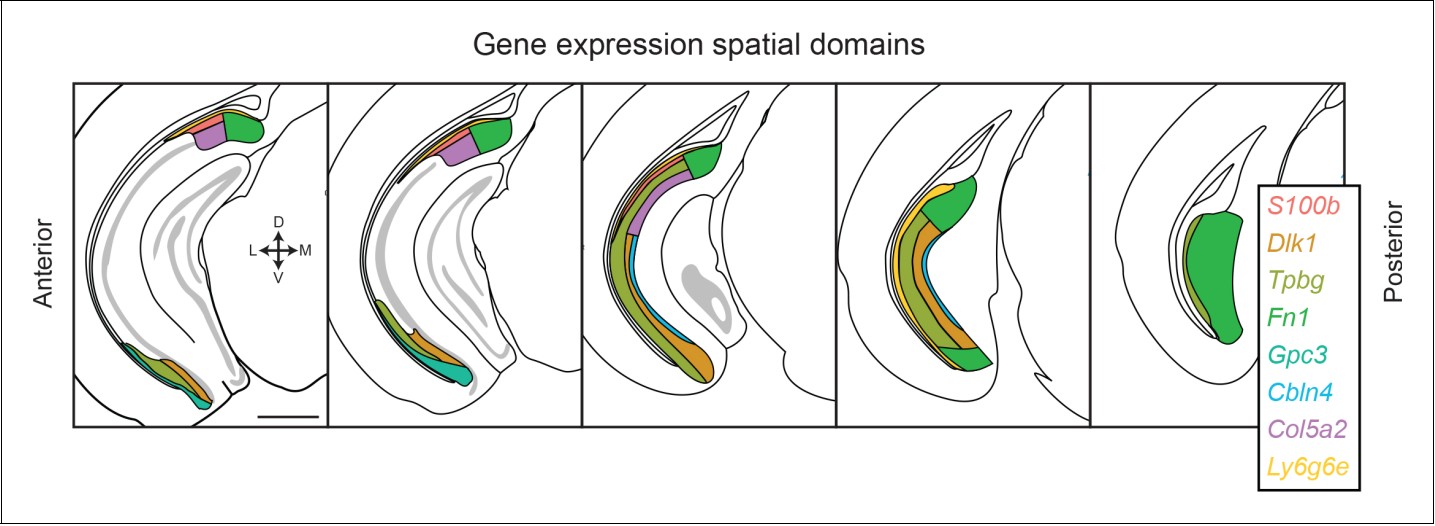

**Figure 7.** Transcriptomic landscape of the subiculum. Schematized spatial domains are illustrated for scRNA-seq clusters across the anterior-posterior axis of the subiculum. The subiculum contains transcriptomically heterogeneous subclasses that conform to a complex geometry. Note that coloring convention for *Ly6g6e* has been changed relative to other figures to differentiate this subclass from the *S100b*-expressing subclass. Scale bar: 1 mm.
DOI: https://doi.org/10.7554/eLife.37701.015

accurately predicted subpopulation-specific organization in entirely separate animals (*Figure 2—figure supplement 3D*).

## Spatial deconstruction of the subiculum

The clusters of excitatory neurons likely reflect different subclasses of subiculum pyramidal cells but may also include cell types from neighboring regions (e.g. CA1). To examine the extent to which these clusters corresponded to subiculum subclasses, we next sought to identify spatial patterns associated with each cluster. We identified cluster-specific marker genes for which Allen Mouse Brain Atlas coronal *in situ* hybridization (ISH) (*Lein et al., 2007*) images were available, and examined the spatial expression of these marker genes (*Figure 3*). The marker genes *Gpc3*, *Dlk1*, and *Tpbg* were strongly expressed ventrally in anterior sections, with *Dlk1* and *Tpbg* exhibiting dorsal expression in more posterior sections. Alternatively, *Col5a2* and *S100b* were expressed in disparate populations of dorsal proximal subiculum (i.e., close to CA1), whereas *Fn1* was enriched in distal subiculum (i.e. away from CA1) (*Cembrowski et al., 2018*) (note that in coronal sections, distal subiculum is primarily associated with enrichment in posterior sections; see *Figure 3—figure supplement 1*). The gene *Ly6g6e* labeled the deepest layer of cells across the long axis, and *Cbln4* corresponding to a layer of cells in the posterior subiculum. Thus, each of these marker genes corresponded to a continuous spatial subregion of the subiculum (*Figure 3*).

Conversely, expression of *Myo5b* was enriched in a densely packed group of cells proximal to the CA1/subiculum border (*Figure 3*, bottom row). Due to the relatively tight cell body packing associated with this label, we postulated that this *Myo5b* expression might correspond to CA1 pyramidal cells. Consistent with this, expression was seen in more anterior regions of CA1, and CA1 expression of *Myo5b* was identified in previous RNA-seq datasets (*Cembrowski et al., 2016b*) (*Figure 3—figure supplement 2*). Thus, the cluster of cells associated with *Myo5b* expression likely belonged to CA1 pyramidal cells. Importantly, no other clusters exhibited markers associated with off-target gene expression (e.g. inhibitory neurons, *Figure 2B*; pre-, para-, or postsubiculum, *Figure 3—figure supplement 3*).

Previous work has demonstrated that the subPCs can be subdivided into distinct subregions based upon immunohistochemical (IHC) labeling (*Ishihara and Fukuda, 2016*). Specifically, it was shown that ZnT3, Nos, and Pcp4 (encoded by *Slc30a3*, *Nos1*, and *Pcp4*, respectively) all conformed to specific proximal laminae, whereas Vglut2 (encoding by *Slc17a6*) corresponded to the distal subiculum. We verified that expression of these genes corresponded to specific subclasses and obeyed

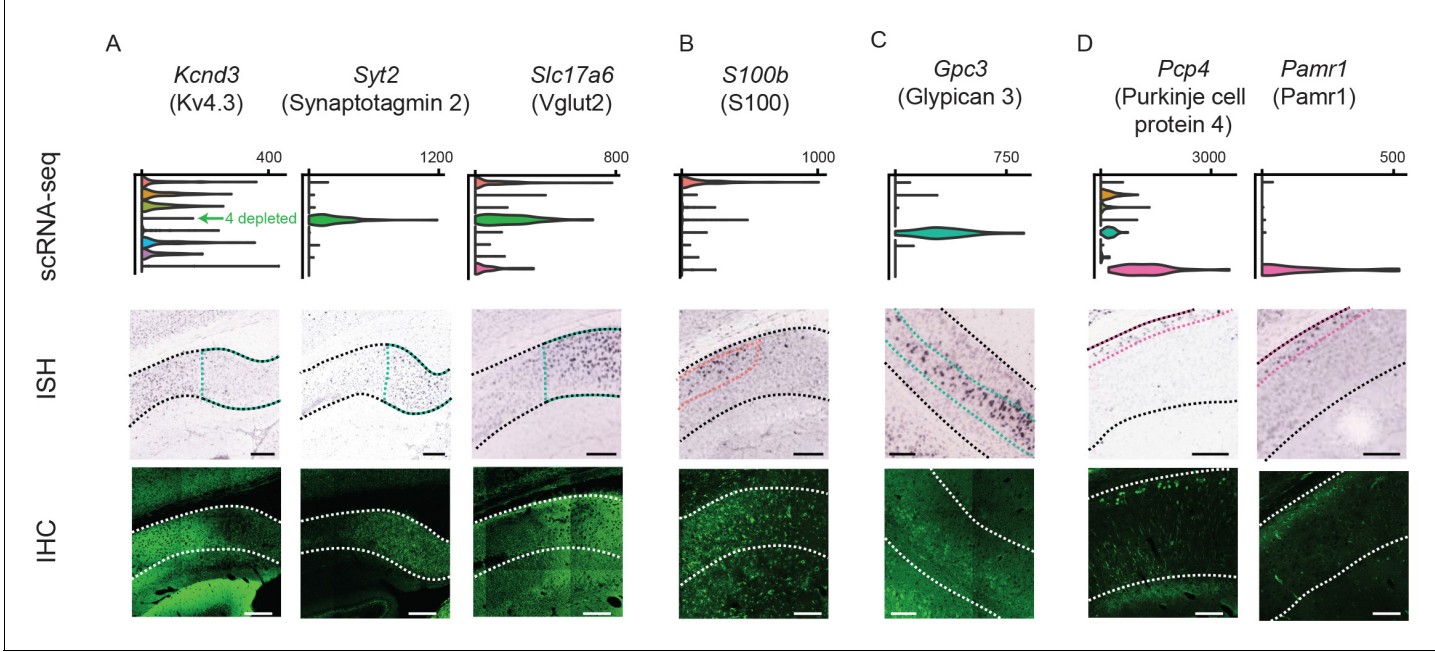

**Figure 8.** Differentially expressed genes correspond to cluster-specific protein products. Top row: gene names, along with associated protein products targeted in IHC. Second row: violin plots of genes that were enriched or depleted in specific clusters. Third row: ISH images of corresponding genes. Black dashed lines illustrate extent of pyramidal cell layer. Colored dashed lines denote spatial domain of associated cluster. Fourth row: immunohistochemical detection of protein products. White dashed lines illustrate extent of pyramidal cell layer. (A) Gene products enriched or depleted in the *Fn1*-expressing cluster (i.e. cluster 4); namely, *Kcnd3* (encoding the potassium channel subunit Kv4.3), *Syt2* (encoding synaptotagmin 2, involved in exocytosis), and *Slc17a6* (encoding Vglut2, mediating glutamate uptake into synaptic vesicles). (B) Results for S100, expressed in the *S100b*-expressing cluster (i.e. cluster 1). Note that the antibody recognizes S100 (i.e. both S100B and S100A) and thus labels astrocytes as well as neurons. (C) Results for the gene product *Gpc3*/Gpc3, enriched in the *Gpc3*-expressing cluster (i.e. cluster 5). (D) Results for the gene products *Pcp4*/Pcp4 and *Pamr1*/Pamr1, enriched in the *Ly6g6e*-expressing cluster (i.e. cluster 9). All scale bars: 200 μm.

DOI: https://doi.org/10.7554/eLife.37701.016

The following figure supplements are available for figure 8:

**Figure supplement 1.** scRNA-seq results guide identification of cluster-specific transgenic mouse line.
DOI: https://doi.org/10.7554/eLife.37701.017
**Figure supplement 2.** S100 protein is present in subiculum pyramidal cell bodies and primary dendrites in the proximal subiculum.
DOI: https://doi.org/10.7554/eLife.37701.018

the spatial organization expected by IHC (*Figure 3—figure supplement 4*). Thus, our work recapitulated and extended this previous work by providing whole-genome and quantitative validation into putative subclasses of subPCs, as well as revealing a host of previously unresolved subclasses.

## Laminar differences in subiculum identity

Given that we were able to spatially register expression of subPC marker genes across the subiculum, we next investigated these spatial domains in finer detail. We began by studying gene expression associated with subiculum laminae. Inspecting the dorsal subiculum first, we found that *Col5a2*, *Tpbg*, and *Ly6g6e* seemingly corresponded to three distinct laminae, patterning the subiculum in a superficial-to-deep fashion (*Figure 4A*). Similarly, in the ventral subiculum the combination of *Cbln4*, *Dlk1*, *Tpbg*, and *Ly6g6e* defined a laminar subiculum organization (*Figure 4B*).

We sought to directly confirm that this lamina-like organization corresponded to mutually exclusive groups of cells, rather than adhering to continua (*Cembrowski and Menon, 2018a*). Using two-color ISH, we labeled for the expression of marker genes *Tbpg* and *Gpc3*, two genes that were mutually exclusive in scRNA-seq (*Figure 3*) and seemingly corresponded to abutting lamina in single-color ISH (*Figure 4B*). Using this strategy, we verified that these laminae corresponded to distinct, abutting but non-overlapping populations of cells (99% of 772 labeled cells exhibited mutual exclusion, n = 2 mice, two sections/mouse; *Figure 5A*). Similar reciprocal laminar organization could

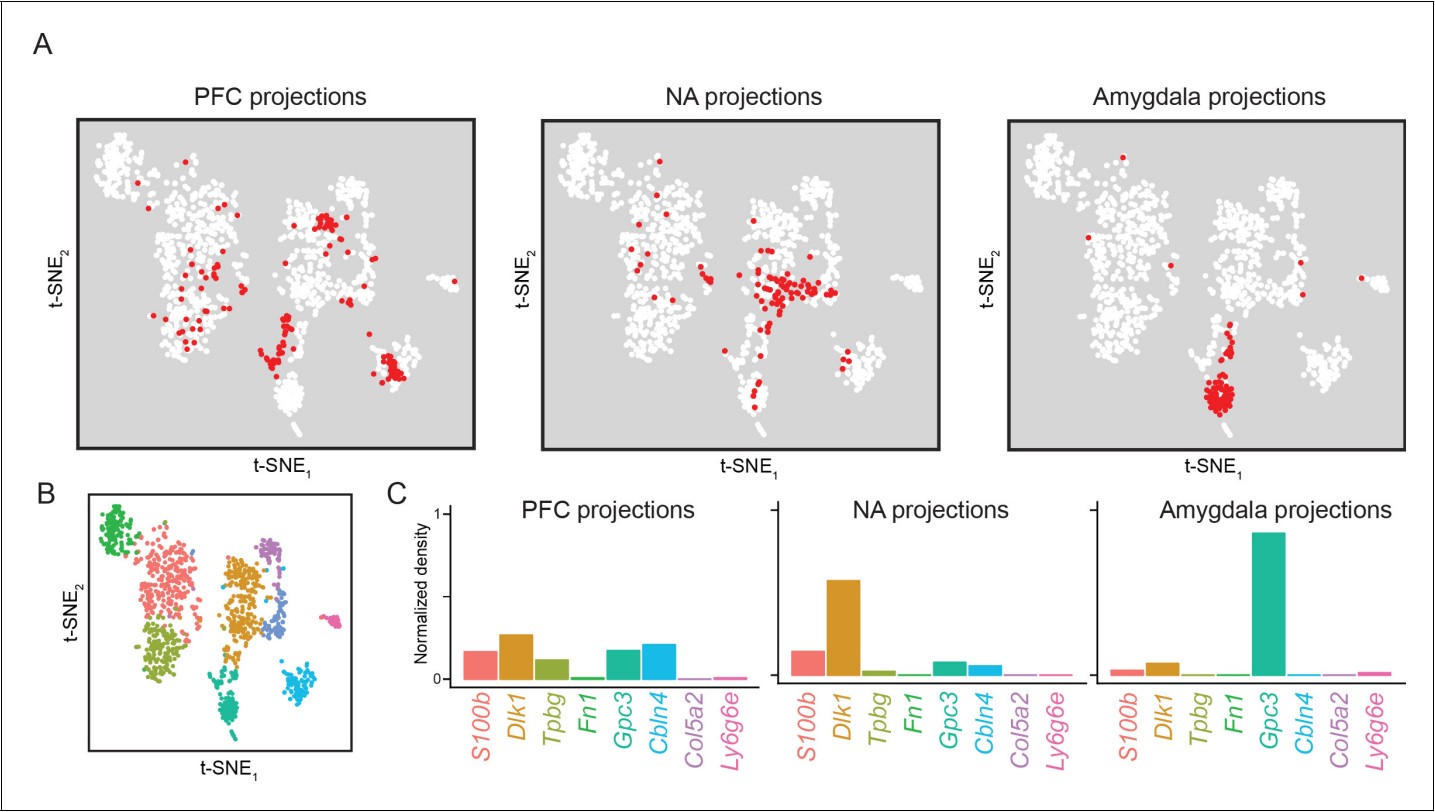

**Figure 9.** Subiculum transcriptomes based upon downstream projections. (**A**) Cells corresponding to three downstream projections (prefrontal cortex, 'PFC'; nucleus accumbens, 'NA'; amygdala) are highlighted (red). (**B**) t-SNE plot of single-cell transcriptomes, illustrating cluster identity (as in *Figure 2A*). (**C**) Relative occupancy for each of the transcriptomic clusters, defined as the number of cluster-specific cells divided by the total number of projection cells, is shown for each projection class.

DOI: https://doi.org/10.7554/eLife.37701.019

The following figure supplements are available for figure 9:

**Figure supplement 1.** Projection-specific transcriptomes associated with replicate dataset.
DOI: https://doi.org/10.7554/eLife.37701.020

**Figure supplement 2.** NA-projecting cells are associated with a different lamina than *Tpbg*-expressing cells.
DOI: https://doi.org/10.7554/eLife.37701.021

**Figure supplement 3.** Spatial representation of projection classes.
DOI: https://doi.org/10.7554/eLife.37701.022

be identified for additional marker genes and associated subclasses (*Fn1* vs. *Dlk1*: 98% of 457 labeled cells exhibited mutual exclusion, *Figure 5B*; *Tpbg* vs. *Dlk1*: 97% of 801 labeled cells exhibited mutual exclusion, *Figure 5C*; *Cbln4* vs. *Dlk1*: 98% of 489 labeled cells exhibited mutual exclusion, *Figure 5D*; *Tpbg* vs. *Ly6g6e*: 93% of 733 labeled cells exhibited mutual exclusion, *Figure 5E*; all statistics represent results from n = 2 mice, two sections/mouse). In total, these findings illustrated the discretely separated nature of multiple scRNA-seq clusters, and revealed that the subiculum exhibited abutting laminae that corresponded to transcriptomically distinct subclasses.

## Most transcriptomic subclasses span the long axis

As long-axis heterogeneity may underlie the complex functionality of the hippocampus (*Strange et al., 2014*), we next considered whether transcriptomic cell classes traversed the long axis (*Cembrowski et al., 2016a*; *Cembrowski et al., 2016b*; *Thompson et al., 2008*). For each cluster, we identified the dorsal and ventral extremes of associated marker gene expression. This analysis revealed that most clusters (6/8) exhibited marker gene expression that traversed most or all of the hippocampal long axis (*Figure 6*, top). The only exceptions to this rule were transcriptomic clusters associated with *S100b* and *Gpc3*, which respectively spanned the dorsal and ventral halves of

the subiculum (*Figure 6*, bottom). Thus, in total, the transcriptomically identified subclasses of subPCs produced a complex geometry that exhibited heterogeneity in the dorsal-ventral, superficial-deep, and proximal-distal axes (*Figure 7*; see also *Figure 3—figure supplement 1*). However, given that most subclasses traversed the dorsal-ventral axis, the primary axes of variation were superficial-deep and proximal-distal (*Witter, 2006*).

## Higher order correlates of transcriptomic clusters

Having combined scRNA-seq and ISH to deconstruct the transcriptomic landscape of the subiculum, we next sought to understand to what extent transcriptomic subclasses covaried with higher-order properties. First, using immunohistochemistry (IHC), we examined to what extent transcript-level differences corresponded to differential protein products. We found that Kv4.3, a potassium channel pore-forming subunit encoded by *Kcnd3*, was depleted in distal subiculum (i.e. the *Fn1*-associated cluster 4) (*Figure 8A*, left). Conversely, this region was enriched for synaptotagmin-2 (a calcium sensor that mediates vesicular release, encoding by *Syt2*) and Vglut2 (a glutamate transporter encoded by *Slc17a6*) (*Figure 8A*, middle and right) (*Ishihara and Fukuda, 2016*). Interestingly, we also found that the *Slc17a6* expression could be exploited for subclass-specific access of the distal subiculum in transgenic mice (*Vong et al., 2011*), providing direct evidence that our transcriptomic work can be leveraged to target and manipulate specific cell types (*Figure 8—figure supplement 1*) (see also *Cembrowski et al., 2018*; *Yamawaki et al., 2018*).

Individual marker genes were also sufficient to delineate protein products in other clusters. Interestingly, the calcium peptide S100, typically used as an astrocyte marker, was found in dendrites and cell bodies of the cluster associated with marker gene *S100b* (*Figure 8B*; see *Figure 8—figure supplement 2* for expansion). Glypican 3, encoded by *Gpc3*, was located ventrally and corresponded to a specific lamina (*Figure 8C*). The proteins Purkinje Cell Protein 4 (*Ishihara and Fukuda, 2016*) and Pamr1, both associated with markers of deep subiculum neurons, exhibited deep lamina-specific enrichment. In total, our scRNA-seq dataset identified multiple spatially restricted proteins, including many important for neuronal functionality (e.g. intrinsic excitability, calcium handling, synaptic transmission).

Finally, we specifically examined our datasets that were obtained based upon projection targets (*Figure 1*). These datasets included three projection targets: the prefrontal cortex (PFC), nucleus accumbens (NA), and amygdala. Each of these datasets represent projection-specific fluorophore-tagged cells that were selectively obtained by manual selection (see Materials and methods). Cells from these projections were differentially distributed across transcriptomic clusters (*Figure 9*; see also *Figure 9—figure supplement 1* for replicate cross-validation). Broadly, both PFC and NA projections tended to be relatively diffusely spread across clusters, although each projection was notably absent from specific subpopulations (e.g. NA projections were *Tpbg*-negative; see *Figure 9—figure supplement 2*). In contrast, amygdala-projecting cells were largely associated with a single dedicated transcriptomic subclass (86% of amygdala-projecting cells were within the *Gpc3* cluster, relative to 8% and 17% of NA- and PFC-projecting cells). Some transcriptomic subpopulations were completely devoid of projections for all surveyed downstream regions (e.g. *Ly6g6e* and *Col5a2* clusters; see *Figure 9—figure supplement 3* for overview), suggesting that they may correspond to other extrahippocampal projections (*Cembrowski et al., 2018*) and/or function as local excitatory neurons (*Xu et al., 2016*).

## Discussion

In this study, we examined the organizational rules underlying heterogeneity within the pyramidal cell population of the subiculum. Using scRNA-seq, we identified widespread differential expression of genes within this canonical neuronal type, and mapped this heterogeneity onto specific subclasses of cells. Using *in situ* hybridization, we identified that these subclasses exhibited mutually exclusive, abutting spatial domains within the subiculum. Furthermore, we found that these transcriptomic classes correlate with protein products and downstream projection targets. Thus, the subiculum can be deconstructed into subfields of principal cells that covary in multiple properties. We have publicly hosted these scRNA-seq data, in conjunction with analysis and visualization tools, to facilitate further study of gene expression and cell types within the subiculum.

## The subiculum as a laminar and columnar structure

From previous cellular- and circuit-level studies, different conclusions have been reached as to the ultimate spatial organization of the subiculum. In one study employing immunohistochemistry, it was demonstrated that several proteins exhibit different laminar-like spatial domains within the subiculum that covary with morphological differences (*Ishihara and Fukuda, 2016*). These findings are suggestive of a laminar organization being present in the subiculum; however, it is challenging to extrapolate governing organizational schemes based on the patterning of a select few markers. In addition, such marker-based approaches do not resolve the overall extent of heterogeneity between putative subclasses, nor guarantee that all potential subclasses are resolved.

Reinforcing the laminar nature of subiculum heterogeneity, complementary circuit-tracing experiments have demonstrated superficial-deep differences in axonal projections (*Ishizuka, 2001*; *Witter, 2006*). Interestingly, such work has also revealed heterogeneity the proximal-distal axis, which is recapitulated by differences in electrophysiological properties (*Cembrowski et al., 2018*; *Jarsky et al., 2008*; *Kim and Spruston, 2012*). In combination, this previous work demonstrates that multiple organizational schemes may be present in the subiculum (laminar and columnar differences), but it is unclear to what extent they can be rectified and ultimately interpreted according to distinct subclasses of cells.

The scRNA-seq approach used here, providing an unbiased and complete (i.e. whole genome) method of assessing a feature of the nervous system, illustrates that both laminar and columnar organizational schemes are simultaneously present and reflect intrinsically heterogeneous subclasses of pyramidal cells. In the proximal subiculum (e.g. proximal to *Fn1*-expressing cells, which define the most distal subclass; see *Figure 3—figure supplement 1*), transcriptomically discrete subclasses of cells occupy abutting laminae (*Figures 5* and *7*). In the distal subiculum, gene expression tends to be relatively homogeneous (although note a lamina of *Tpbg*-expressing cells in posterior subiculum: *Figures 3* and *7*). Thus, the subiculum can be deconstructed into proximal and distal subdomains, with further laminar organization predominantly found in proximal subiculum (as previously proposed by *Ishihara and Fukuda, 2016*).

To what extent does our work unambiguously resolve the subclass-specific landscape of the subiculum? Here, we directly demonstrated the discrete pairwise separation of five clusters (*Figure 5*), and the overall nonoverlapping nature of these clusters can be inferred from their relative spatial ordering. Taking these results in combination with previous *in situ* hybridization (*Cembrowski et al., 2018*), this demonstrates the existence of at least six discretely separable subclasses of subiculum pyramidal cells. Although not examined directly, it is possible that some remaining scRNA-seq clusters may comprise opposite extremes of a continua. On the other hand, there may be additional subiculum subclasses that may be revealed with greater cell number or sequencing depth. As a result, the eight scRNA-seq subclasses resolved here likely represent an approximation (and potentially, a lower bound) as to the ultimate number of true biological subclasses of subiculum pyramidal cells.

## Transcriptomic heterogeneity as a predictor of functional heterogeneity

Understanding how heterogeneity within the hippocampus underpins function has conventionally been studied by comparing across classical hippocampal cell types (e.g. *Kaifosh and Losonczy, 2016*; *Neunuebel and Knierim, 2014*). Complementing this body of across-cell-type work, recent transcriptomic research has illustrated that heterogeneity within each classical hippocampal cell type is also prominent. This heterogeneity encompasses both discrete and continuous variation across dorsal-ventral, proximal-distal, and superficial-deep axes (*Cembrowski et al., 2016a*; *Cembrowski et al., 2018*; *Cembrowski et al., 2016b*; *Habib et al., 2016*; *Thompson et al., 2008*). As higher order cellular, circuit, and functional features also vary in related ways (*Cembrowski and Spruston, 2018b*, in review; *Danielson et al., 2016*; *Knierim et al., 2006*; *Lee et al., 2015*; *Lee et al., 2014*; *Soltesz and Losonczy, 2018*; *Strange et al., 2014*), this suggests that transcriptomic identity can be coherently aligned with specialized functionality (*Cembrowski et al., 2018*; *Yamawaki et al., 2018*).

It follows that the transcriptomically defined subclasses identified in this study likely vary according to higher-order structure and function. This postulate is further underscored by several

complementary lines of evidence. For example, there is widespread differential expression of genes associated with neuronally relevant ontologies (*Figure 2C*). Additionally, in the case of broadly defined proximal-distal cell classes in the dorsal subiculum, dissociable higher order structural and functional correlates have been previously identified (*Cembrowski et al., 2018*). Finally, for several of the transcriptomic classes identified in this study, transcriptomic identity covaries with protein products (*Figure 8*) and projection target (*Figure 9*). In combination, these lines of evidence indicate that the transcriptomic classes identified here correspond to functionally differentiable and relevant subclasses of subPCs.

How can such function be identified? In this study, we identified that subPCs can be deconstructed into a collection of discretely separated subclasses based upon disparate gene expression. This approach exploited gene expression heterogeneity as a means of cellular classification and spatial registration. As a consequence, this analysis was performed agnostic to the functional correlates of these genes; however, this work will help to provide a necessary foundation for assessing functional relevance in multiple ways. First, many of the differentially expressed genes in this study are associated with known functional roles in neuronal populations (*Figure 2C*). Consequently, these findings enable specific hypotheses to be generated and tested across subclasses. Second, as these subclasses can covary with projection target (*Figure 9*), these predictions can be investigated and understood at the level of neuronal circuits. Third, our analysis provides individual genes as markers (*Figures 2* and *3*) that will enable these questions to be addressed at a subclass-specific resolution (e.g. via transgenic mice; *Figure 8—figure supplement 1*). In total, this work will facilitate the coherent interrogation of molecular, cellular, and circuit properties of the specific subclasses of the subiculum.

## Single-cell Hipposeq, a public resource for hippocampal scRNA-seq

Due to the data-rich nature of our subiculum scRNA-seq dataset, there are many additional features that can be mined and analyzed in further studies. To facilitate the extended use of these data, we have publicly hosted our scRNA-seq data in conjunction with corresponding analysis and visualization tools. This augments earlier population-level RNA-seq data hosted by our laboratory ('Hipposeq': *Cembrowski et al., 2016b*), providing an accessible and intuitive single-cell extension for dissecting the structural and functional heterogeneity of the subiculum. Thus, our work here provides both an immediate and long-term framework with which subiculum subclasses can be interpreted, targeted, and manipulated in future studies.

# Materials and methods

**Key resources table**

| Reagent type (species) or resource | Designation | Source or reference | Identifiers | Additional information |
|---|---|---|---|---|
| Strain, strain background (*M. musculus*) | Vglut2-IRES-Cre | Jackson | RRID: IMSR_JAX: 016963 | |
| Antibody | Kv4.3 rabbit polyclonal | Alomone | APC-017; RRID: AB_2040178 | 1:200 |
| Antibody | Syt2 mouse monoclonal | DSHB | RRID: AB_531910 | 1:250 |
| Antibody | Vglut2 mouse monoclonal | Abcam | ab79157, RRID: AB_1603114 | 1:1000 |
| Antibody | S100 rabbit polyclonal | Abcam | ab868, RRID: AB_1603114 | 1:250 |
| Antibody | Gpc3 mouse monoclonal | Millipore | MABC667 | 1:250 |

*Continued on next page*

*Continued*

| Reagent type (species) or resource | Designation | Source or reference | Identifiers | Additional information |
|---|---|---|---|---|
| Antibody | Pcp4 rabbit polyclonal | Sigma | HPA005792, RRID: AB_1855086 | 1:250 |
| Antibody | Pamr1 rabbit polyclonal | Proteintech | 55310–1-AP, RRID: AB_11232 | 1:250 |
| Sequence-based reagent | Tpbg ISH probe | Advanced Cell Diagnostics | 521061-C3 | |
| Sequence-based reagent | Dlk1 ISH probe | Advanced Cell Diagnostics | 405971-C2 | |
| Sequence-based reagent | Gpc3 ISH probe | Advanced Cell Diagnostics | 418541 | |
| Sequence-based reagent | Fn1 ISH probe | Advanced Cell Diagnostics | 310311 | |
| Sequence-based reagent | Cbln4 ISH probe | Advanced Cell Diagnostics | 428471 | |
| Sequence-based reagent | Ly6g6e ISH probe | Advanced Cell Diagnostics | 506391-C2 | |
| Software, algorithm | R | https://www.r-project.org | SCR_001905 | |
| Software, algorithm | Seurat | https://satijalab.org/seurat/ | SCR_007322 | |
| Software, algorithm | Fiji | https://imagej.net/Fiji | RRID: SCR_002285 | |
| Software, algorithm | Custom scripts | This study | DOI: 10.6084/m9.figshare.7140350 | Scripts used to analyze scRNA-seq data |
| Other | Retrobeads | Lumafluor | | '*Overview of subiculum scRNA-seq atlas: construction, validation, and extension*' |
| Other | AAV-SL1-CAG-tdT | Janelia Viral Core | | '*Higher-order correlates of transcriptomic clusters*' |

Experimental procedures were approved by the Institutional Animal Care and Use Committee at the Janelia Research Campus (protocols 14–118 and 17–159). Mice were housed on a 12 hr light/dark cycle with ad libitum food and water access.

## scRNA-seq data generation and analysis

An initial single-cell RNA-seq dataset (5.6 ± 1.0 thousand expressed genes/cell, mean ± SD) was generated according to a previously published protocol (*Cembrowski et al., 2018*). In brief, for animals used in geography-based datasets (dorsal, intermediate, and ventral), mature (>8 weeks) male C57BL/6 mice were used. In these animals, coronal sections were made, and microdissection of the corresponding geographical regions was performed (n = 1 biological replicate, that is animal, for each region). Microdissected regions were dissociated, and manual purification (*Hempel et al., 2007*) was used to obtain cells. For animals used in projection-based datasets (PFC, NA, and

amygdala; n = 1 biological replicate, that is animal, for each region), red or green retrograde beads (Lumafluor, Naples, FL) were injected bilaterally at 200 nL/depth as follows: PFC: A/P, M/L, D/V 2.0, 0.25, (-2.5,–2.25); NA: 2.0, 1.0, (-5.0,–3.8); amygdala: −0.5, 2.8, (-5.0,–4.0). One injection site along the anterior-posterior axis was selected for each site to avoid potential off-target effects associated with injecting large volumes of the brain. Fluorescent cells in the intermediate and ventral subiculum were targeted for manual purification according to previous methods (*Cembrowski et al., 2016a*), with 175, 139, and 71 cells obtained for the NA, PFC, and amygdala, respectively. To validate this initial scRNA-seq dataset, a second scRNA-seq was constructed and analyzed independently (n = 884 cells, with $5.6 \pm 0.8$ thousand genes expressed/cell). This dataset contained unlabeled cells selected at random across the full extent of the subiculum (n = 2 biological replicates; i.e., mice), as well as projection-specific datasets (n = 2, 1, and one biological replicates from the NA, PFC, and amygdala, respectively, with n = 116, 64, and 44 labeled projection cells obtained for each respective projection).

For all datasets, library preparation, sequencing, and initial count-based quantification (*Dobin et al., 2013*; *Trapnell et al., 2009*) was performed according to previous methods (*Cembrowski et al., 2018*); note that the dorsal subiculum dataset was previously published and publicly available as part of this earlier work. For some datasets, barcodes that could not be demultiplexed were mapped to known barcodes using maximally parsimonious substitutions. No blinding or randomization was used for the construction or analysis of this dataset. No *a priori* sample size was determined for the number of animals or cells to use; note that previous methods have indicated that several hundred cells from a single animal is sufficient to resolve heterogeneity within the subiculum (*Cembrowski et al., 2018*).

Computational analysis was performed in R (RRID:SCR_001905) (*R Development Core Team, 2008*) using a combination of Seurat (RRID:SCR_007322) (*Satija et al., 2015*) and custom scripts (*Cembrowski et al., 2018*). Cells with <10,000 total counts were excluded from analysis (n = 60 of 1190 initial cells). For all remaining cells, counts were converted to Counts Per Million (CPM) for subsequent analysis. Putative non-neuronal cells (n = 14) were eliminated from the dataset by rejecting cells that exhibited CPM < 250 for *Snap25*, a pan-neuronal marker. Putative interneurons (n = 13) were eliminated from the dataset by rejecting cells that exhibited CPM > 100 for *Gad1*, an interneuron marker. Variable genes (n = 5376) used for PCA were obtained with Seurat via *FindVariableGenes(mean.function = ExpMean, dispersion.function = LogVMR, x.low.cutoff = 0.0125, x.high.cutoff = 3, y.cutoff = 0.5)*. Clusters were obtained with Seurat via *FindClusters(reduction.type = 'pca', dims.use = 1:10, resolution = 0.6)*. In general, these parameters produced clusters that were robust (e.g. *Figure 2—figure supplement 1b*) and cross-validated by other methodologies (e.g. *Figures 3*, *4*, *5*, *7*, *8* and *9*) (*Cembrowski and Spruston, 2017*). This requirement of multimodal consistency produces a conservative but well-validated approach to identify subclasses. Hierarchical clustering of clusters was obtained with Seurat via *BuildClusterTree()*. Subclass-specific enriched genes (*Figure 2—figure supplement 2*) were obtained with Seurat via *FindMarkers()*, retaining genes that were at least 3-fold enriched in the target population (the 'enriched cluster', relative to the 'depleted cluster') and obeyed $p_{ADJ} < 0.05$, where is the $p_{ADJ}$ is adjusted $p$ value from Seurat based on Bonferroni correction. Functionally relevant differentially expressed genes (*Figure 2C*) were obtained using *FindMarkers()*, allowing for both cluster-specific enriched and depleted genes obeying $p_{ADJ} < 0.05$. t-SNE visualization (*van der Maaten and Hinton, 2008*) used perplexity = 30, with 1000 iterations (sufficient for convergence) on the default seed. Qualitatively similar results were obtained for other seed values.

When plotting gene expression using t-SNE, color ranges from white (zero expression) to red (maximal expression), plotted logarithmically. For random forest classification (*ClassifyCells()* in Seurat), random subsets of graph-based clustered cells were taken (n = 50, 100, 200, 400, or 800 cells; n = 100 random subsets for each number of cells), and used to predict the cluster identities of the remaining cells in the dataset.

Raw and processed scRNA-seq datasets have been deposited in the National Center for Biotechnology Information (NCBI) Gene Expression Omnibus under GEO: GSE113069. All analysis scripts are publicly available (DOI: 10.6084/m9.figshare.7140350) (*Cembrowski and Spruston, 2018c*).

## *In situ* hybridization

All chromagenic ISH images were obtained from the publicly available Allen Mouse Brain Atlas (AMBA) (*Lein et al., 2007*) (*Supplementary file 3*). To cross-validate marker genes associated with scRNA-seq clusters, we identified AMBA coronal image sets for genes that exhibited minimal off-target expression in scRNA-seq datasets. To cross-validate expression of *Myo5b* in CA1 cells in RNA-seq, previous population-level RNA-seq was used (*Cembrowski et al., 2016b*).

All multicolor fluorescent ISH processing was performed according to previous protocols (*Cembrowski et al., 2016a*). All probes were purchased from Advanced Cell Diagnostics (Hayward, CA) and were as follows: *Tpbg* (521061-C3), *Dlk1* (405971-C2), *Gpc3* (418541), *Fn1* (310311), *Cbln4* (428471), *Ly6g6e* (506391-C2). For combining ISH with circuit mapping, AAV-SL1-CAG-tdTomato (rAAV2-retro: *Tervo et al., 2016*) was injected into the NA, with the same coordinates used in retrobead injections (200 nL/site; note that retrobeads were not used due to bead labeling being lost during ISH processing). For quantifying colocalization of two-color ISH, cell bodies were counted across at least two optical sections from two animals, with the degree of overlap quantified as the number of colabeled cells divided by the total number of labeled cells in either channel.

## Immunohistochemical and transgenic mouse validation

Male mice (>=2 mice/antibody) were deeply anesthetized with isoflurane and perfused with 0.1M phosphate buffer (PB) followed by 4% paraformaldehyde (PFA) in PB. Brains were dissected and post-fixed in 4% PFA overnight. For most experiments, brain sections (50 – 100 μm) were made using a vibrating tissue slicer (Leica VT 1200S, Leica Microsystems, Wetzlar, Germany; where noted, some experiments used cryostat-sectioned tissue (Leica 3050S, Leica Microsystems, Wetzlar, Germany). Antibodies used in this study were as follows: on rabbit antibody to Kv4.3 (1:200, APC-017, Alomone; RRID: AB_2040178), mouse antibody to Syt2 (1:250, znp-1, DSHB; RRID: AB_531910; performed on cryosectioned tissue), mouse antibody to Vglut2 (1:1000, ab79157, Abcam; RRID: AB_1603114), rabbit antibody to S100 (1:250, ab868, Abcam; RRID: AB_1603114), mouse antibody to Gpc3 (1:250, MABC667, Millipore), rabbit antibody to Pcp4 (1:250, HPA005792, Sigma; RRID: AB_1855086), rabbit antibody to Pamr1 (1:250, 55310–1-AP, Proteintech; RRID: AB_11232034).

Immunohistochemistry was performed on free-floating sections. All tissues were washed five times (5 min each) in PBS and then incubated in blocking buffer (5% NGS in 0.3% Triton-PBS; Kv4.3 and Vglut2 IHC additionally used 2% BSA) for 1 hr at room temperature. Tissue was subsequently incubated in primary antibody at 4°C for one to two nights, washed five times (5 min each) in 0.3% Triton-PBS, and detected by Alexa Fluor secondary antibodies (Thermo Scientific Inc., Waltham, MA) by incubating at room temperature for 1 – 2 hr. Sections were subsequently washed in PBS five times (5 min each), mounted, and coverslipped with mounting media containing DAPI (H-1200, Vector Laboratories, Burlingame, CA).

For investigating cell-type-specific access predicted by scRNA-seq, Slc17a6-IRES-cre (i.e. Vglut2-IRES-cre; RRID: IMSR_JAX:016963) (*Vong et al., 2011*) male mice (n = 4) were injected with AAV2/1-CAG-FLEX-EGFP (Janelia Virus Services) in the subiculum (A/P −3.6, M/L 2.5; D/V −2.5 and −1.5 with 80 nL/depth). Mice were sacrificed at least 3 weeks later for histological examination of viral expression.

## Fluorescence imaging

All histological images were acquired with a 20x objective using confocal microscopy (LSM 880, Carl Zeiss Microscopy, Jena, Germany). Single optical sections are shown, with the relevant regions tiled in XY dimensions as needed. In some cases, channels were postprocessed in Fiji (RRID:SCR_002285) (*Schindelin et al., 2012*), with brightness adjustments applied to the entire image and/or pseudocoloring.

# Additional information

## Funding

| Funder | Author |
| --- | --- |
| Howard Hughes Medical Institute | Mark S Cembrowski<br>Lihua Wang<br>Andrew L Lemire<br>Monique Copeland<br>Salvatore F DiLisio<br>Jody Clements<br>Nelson Spruston |

The funders had no role in study design, data collection and interpretation, or the decision to submit the work for publication.

## Author contributions

Mark S Cembrowski, Conceptualization, Data curation, Software, Formal analysis, Validation, Investigation, Visualization, Methodology, Writing—original draft, Writing—review and editing; Lihua Wang, Monique Copeland, Resources, Validation, Investigation, Methodology, Writing—review and editing; Andrew L Lemire, Conceptualization, Resources, Data curation, Investigation, Methodology, Writing—review and editing; Salvatore F DiLisio, Investigation, Methodology, Writing—review and editing; Jody Clements, Software, Visualization, Writing—review and editing; Nelson Spruston, Conceptualization, Supervision, Funding acquisition, Project administration, Writing—review and editing

## Author ORCIDs

Mark S Cembrowski http://orcid.org/0000-0001-8275-7362
Nelson Spruston http://orcid.org/0000-0003-3118-1636

## Ethics

Animal experimentation: Experimental procedures were approved by the Institutional Animal Care and Use Committee at the Janelia Research Campus (protocols 14-118 and 17-159).

## Decision letter and Author response

Decision letter https://doi.org/10.7554/eLife.37701.036
Author response https://doi.org/10.7554/eLife.37701.037

# Additional files

## Supplementary files

• Supplementary file 1. List of cluster-specific enriched genes. See tableS1.txt.
DOI: https://doi.org/10.7554/eLife.37701.023

• Supplementary file 2. List of cluster-specific enriched genes in pairwise comparisons. See tableS2.txt.
DOI: https://doi.org/10.7554/eLife.37701.024

• Supplementary file 3. List of Allen Mouse Brain Atlas images used in text. See tableS3.txt.
DOI: https://doi.org/10.7554/eLife.37701.025

• Transparent reporting form
DOI: https://doi.org/10.7554/eLife.37701.026

## Data availability

Raw and processed scRNA-seq datasets have been deposited in the National Center for Biotechnology Information (NCBI) Gene Expression Omnibus under GEO: GSE113069.

The following dataset was generated:

**Database and**

| Author(s) | Year | Dataset title | Dataset URL | Identifier |
|-----------|------|---------------|-------------|------------|
| Cembrowski MS | 2018 | The subiculum is a patchwork of discrete subregions | https://www.ncbi.nlm.nih.gov/geo/query/acc.cgi?acc=GSE113069 | NCBI Gene Expression Omnibus, GSE113069 |

The following previously published dataset was used:

| Author(s) | Year | Dataset title | Dataset URL | Database and Identifier |
|-----------|------|---------------|-------------|-------------------------|
| Cembrowski MS | 2018 | Dissociable structural and functional hippocampal outputs via distinct subiculum cell classes | https://www.ncbi.nlm.nih.gov/geo/query/acc.cgi?acc=GSE100449 | NCBI Gene Expression Omnibus, GSE100449 |

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
