## [Decision Letter]

Thank you for sending your article entitled "The subiculum is a patchwork of discrete subregions" for peer review at *eLife*. Your article has been evaluated by three peer reviewers, one of whom is a member of our Board of Reviewing Editors, and the evaluation has been overseen Gary Westbrook as the Senior Editor.

This paper was submitted as a Tools and Resources contribution and thus should not just serve to provide evidence of cell type heterogeneity in the subiculum. The contribution attempts to go further in providing markers that can be used to manipulate specific cell types in the subiculum, thereby potentially providing a valuable resource that would allow experiments that are not currently possible. However, without sufficient validation of the different classes of subiculum cells, other researchers may hesitate to use such a resource. As is apparent in the reviews, all reviewers had concerns about the lack of a sufficient number of biological replicates. We recommend that results should be replicated across multiple animals (i.e., at least 2 animals, preferably 3). Please also address reviewer 2 and 3's concerns about validation of the clusters (e.g., more detailed *in situ* hybridization for multiple target mRNAs in the same section and/or immunostaining). The reviews are provided in their entirety below.

Reviewer #1:

This is an interesting study that reports molecularly and anatomically distinct subgroups of cells in the subiculum, some of which are shown to project to different targets. These findings are important because they will allow for novel experiments in the future that test the functional consequences of manipulations of different of different types of hippocampal output (e.g., test the effects of subiculum projections to amygdala using *Gpc3* as a cell-specific marker). However, there are some points that remain to be clarified.

1) The term "biological replicate" seems misleading because the authors seem to report that cells from only one animal were analyzed for each region. Is this standard to only use one animal for each region? It seems as though this resource would be most significant to other researchers if the reproducibility of these clusters across animals was demonstrated.

2) In the Discussion, the authors state that they were "agnostic to the functional correlates" of the genes that serve to differentiate cell clusters. However, this should perhaps be explained earlier in the text, specifically in the section of the Results that describes Figure 2. Otherwise, readers may naturally wonder whether these gene expression patterns provide insights about functions of different cell groups.

Reviewer #2:

Cembrowski and colleagues have used single cell RNA-seq to profile pyramidal neurons in the subiculum. They discover that there are 8 or 9 discrete cell types that span and tile the dorsal/ventral, proximal/distal, and superficial/deep axis of the subiculum. This is the most comprehensive analysis of the subiculum to date and of high quality and value. The authors have shared the data through addition to http://hipposeq.janelia.org – a searchable database that already includes data from many cell types in the hippocampus and is of significant value to the community. Finally, the authors use retro beads injected into the PFC, NA, and amygdala to identify subiculum pyramidal neuron that project to these regions and performed some preliminary analysis of these subtypes. This study uses methods and analyses previously developed/used by the Spruston lab to perform a comparable analysis of CA1/CA2/CA3/and DG.

1) The text should be clarified as to whether there are 8 or 9 clusters of cell types.

2) Can any of the clusters be validated with immunostaining or transgenic mouse lines?

3) In the final Results section describing projection specific-correlates of transcriptomic clusters, it is difficult to evaluate the strength of the data. How many neurons were labeled/profiled in each projection class? How complete was the coverage of the target area? The Materials and methods section indicates retro beads and AAV-SL1-CAG-tdTomato were used to label projections but in the Results section, it seems like just beads were used. This should be clarified. A visual for how the projection neurons map onto the spatial domains described in Figure 6 and layers in Figure 4, would be useful.

Reviewer #3:

In this study, Cembrowski et al. performed deep sequencing of RNA from manually isolated neurons from the subiculum. They identified clusters of neurons most of which were extended throughout the dorso-ventral span of the subiculum with variation in the antero-posterior (or proximo-distal) axis. The results confirm and extend previous observations of differences between the proximal and distal subiculum. The authors go on to show that projections of subicular neurons to the nucleus accumbens and amygdala might map onto their transcriptional identity.

The potential general interest of the study is that single-cell RNAseq is relatively new and has yet to be applied to the subiculum, which is an interesting brain area because of its roles in memory. The technique is powerful in that it can identify neurons based on transcriptional signatures and when combined with projection and positional identities it can provide a convincing picture of cell diversity in a brain area. The limitations of the present study are that the claimed conceptual advances are over-stated and that the utility of the resource is compromised by a lack of replicates and limited validation.

1) While the marker genes identified will be useful, layer topography and distinctions in the proximo-distal axis of the subiculum were already largely known. For example, Ishihara and Fukuda (Neuroscience, 2016) have provided strong evidence for a layered molecular organization of the subiculum (nicely summarised in Figure 15 of their paper) and already provide good evidence for the main conclusion expressed in the title of the present study. While their study is very briefly mentioned, its convincing, important and original contributions are not sufficiently acknowledged. They should be outlined in the Introduction and the present data should be compared more directly with their results in the Discussion. It is particularly important to clarify whether the protein markers they identify (e.g. NOS, PCP4, ZnT3) are useful markers at the mRNA level based on the present data. It may also be interesting to address whether the subiculum is analogous to the deep layers of the isocortex or neocortex.

2) The major potential contribution of the present study will likely be for groups wanting to investigate how gene expression patterns in each of the identified subclasses relate to neuronal function or signaling cascades. For this, we find a biological replicate of 1 is not sufficient. There is a real concern here that the results are not definitive, and the study may mislead rather than benefit future work. Since the authors invested effort in making the dataset public, biological replicates would enhance the reliability of their study once it goes in the public domain.

3) There seems to be an overlap in expression of the marker genes expressed by neurons projecting to each target area (Figure 7). For example, neurons projecting to the PFC express nearly all genes enhanced in all clusters, which weakens the conclusion that distinct transcriptional identities map onto projection choice. Similarly, NA and amygdala projection neurons express both *Dlk1* and *Gpc3*, albeit at different proportions. Perhaps with more animals sampled, the level of expression of marker genes will appear more even or more different? We suggest to either increase the number of animals in the sequencing assay, or to perform immunostaining / in situs to find out whether neurons projecting to each area are positive for subclass markers.

4) Selected genes in 5 out of the 8 clusters identified are lamina-specific, but some genes even though they belong to separate clusters seem to be expressed in an overlapping manner. For example, *Cbln4* looks like it is in the same lamina as *Dlk1* from single in-situs (Figure 4B). If the authors want to show this gene really marks a distinct class, they need to perform fluorescent ISH for this gene and *Dlk1*. For the same reason, *Col5a* and *Dlk1*, which are depicted as non-overlapping in Figure 6, need validation with double ISH. Also, *Tpbg* is depicted as restricted to the ventral half of the subiculum, even though it appears to be expressed more dorsally (Figure 3). A diagram indicating which genes are expressed in which lamina may also be helpful – this is hard to see in Figure 6.

5) In the Abstract, the claim "the subiculum pyramidal cell population can be deconstructed into eight discretely separable subclasses" is difficult to justify based on the data. There is no test of discreteness in the analysis and the t-SNE plots suggest that the suggested subclasses overlap / are contiguous.

6) Choices made in setting up the analysis pipeline, and the consequences of different choices, are not sufficiently explained. For example, to what extent does changing the cutoff thresholds in FindVariableGenes affect the subsequent clustering? To what extent is clustering affected by the resolution parameter? In generating the t-SNE plots what were the perplexity values and how many steps were used? Several sub-classes do not appear particularly well separated in the t-SNE plots. It may be that optimising perplexity or steps gives a better indication of separation, or it may be that they are not separable even with t-SNE.

7) Figure 4C-E and Figure 7—figure supplement 1 require quantification of the numbers of cells and numbers of biological replicates.

8) Figure 2—figure supplement 1B suggests that approximately 20-25% of cells are not classifiable. Does this population overlap across simulation runs? If so, what happens to the cell classification if these cells are excluded? For example, are the groupings cleaner?

9) For reproducibility, the main analysis code should be made available alongside the study. When doing so it would be helpful to make explicit the random number seeds used by Seurat and the version of Seurat used. When we used Seurat to analyze the dataset we obtained different, although qualitatively similar, plots to those in the manuscript. We guess this is because of differences in seeding but cannot know for sure. The methods should also make clear whether key analyses were repeated with different seeds and whether similar results were obtained.

[Editors' note: further revisions were requested prior to acceptance, as described below.]

Thank you for resubmitting your work entitled "The subiculum is a patchwork of discrete subregions" for further consideration at *eLife*. Your revised article has been favorably evaluated by Gary Westbrook as Senior Editor, a Reviewing Editor, and one reviewer.

Summary:

The manuscript has been improved but there are some remaining issues that need to be addressed before acceptance, as outlined below. Please note that the remaining comments only require text revisions, not additional analyses or experiments. Specifically, we would like a clearer explanation of relevance to prior work and a softening of the conclusion that there are definitely eight separate subregions.

Essential revisions:

1) The revisions do more to acknowledge previous work by Ishihara and Fukuda. However, this previous work is still not credited sufficiently. This doesn't affect the validity of the conclusions, nor the strong case for publishing the study, but a more scholarly approach to recognizing previous work would reflect well on the authors and the journal. We have some suggestions.

Introduction, third paragraph, acknowledges Ishihara and Fukuda, but their contribution is not made sufficiently clear here. Perhaps state, "Recent investigations using immunohistochemical labeling argue that the proximal subiculum is composed of a molecular layer and multiple cell body layers, each distinguished by molecular and morphological differences, while the distal subiculum is more uniform (Ishihara and Fukuda, 2016)."

Discussion, subsection “The subiculum as a laminar and columnar structure”, last paragraph. It's important to be clear here that prior evidence for and articulation of the idea that "the subiculum can be deconstructed into proximal and distal subdomains, with further laminar organization predominantly found in proximal subiculum" was provided in the previous study by Ishihara and Fukuda. We agree that the present results further refine their model, but as written the text could easily be interpreted as taking credit for ideas previously proposed by Ishihara and Fukuda. Also, the implication in the first paragraph of the aforementioned subsection, that Ishihara and Fukuda's scheme was based only on protein labeling is incorrect, as they also took account of neuronal morphology, something the present study doesn't do.

2) We're not convinced by the conclusion that the analysis supports the existence of eight discrete cell classes. It seems pretty clear that some of the cell classes are distinct (e.g. groups labeled by *Cbln4* vs. *Dlk1*), but while others can be grouped, the possibility that rather than being discrete classes they reflect a split along a continuum, or overlapping classes, is not convincingly addressed (e.g. *S100b* vs. *Tpbg*).

First, it's not clear how the algorithm used for clustering provides evidence for groups being generated by discrete as opposed to continuous distributions. As the authors recognize this is a difficult problem. Clustering algorithms will group some continuous distributions into clusters, but this is not evidence that the distributions were in fact discrete. Rather than under-sampling, which doesn't address the problem convincingly, a more rigorous statistical approach is required to support the conclusion that there are eight discrete groups. Previous comments about the interpretation of the tSNE plots must not have been sufficiently clear. They of course don't test for discreteness. Rather, a typical concern is that tSNE can give an exaggerated sense of distances between data points, whereas here even when visualizing the data using tSNE plots the groups appear contiguous.

Second, the new data in Figure 5 are nice, but their interpretation requires clarification. Each comparison is between groups that appear well separated by the clustering algorithm (cf. Figure 2A) and the conclusion that the groups labeled in Figure 5 are distinct seems very reasonable. However, the clusters in Figure 2A that appear contiguous (e.g. *S100b* and *Tpbg, Dlk1* and *Lefty1, Dlk1* and *Gpc3*, etc.) are not validated in Figure 5. Thus, Figure 5 doesn't support the idea that all eight proposed cell groups are discretely separable, as there is no evidence here that the adjacent groups of cells from Figure 2A are separable.

Third, the argument that ~80% successful classification with a random forest validates the groupings is not by itself convincing. For genuinely discrete classes wouldn't you expect the RF to do a little better? Are the misclassified cells randomly distributed, or are errors more systematic? For example, are all groups misclassified at the same rate, or is the error rate higher for particular groups? Are some types of error over-represented, e.g. does *S100b* misclassify as *Tpbg*, and vice -versa, at higher rates than other pairings? What error rates would you expect given alternative hypotheses that some groups, e.g. *S100b* and *Tpbg*, represent a split along a continuum, rather than discrete classes?

These problems could easily be addressed by avoiding use of "discrete" as a conclusion, e.g. in the title, in the Abstract, and multiple other parts of the manuscript, by discussing the potential ambiguities, and by being open to the possibility that some, although clearly not all, of the groupings may not be discrete. Along the same lines, the idea that there are eight groups would be better phrased as a suggestion rather than a definitive conclusion.

---

## [Author Response]

Reviewer #1:This is an interesting study that reports molecularly and anatomically distinct subgroups of cells in the subiculum, some of which are shown to project to different targets. These findings are important because they will allow for novel experiments in the future that test the functional consequences of manipulations of different of different types of hippocampal output (e.g., test the effects of subiculum projections to amygdala using Gpc3 as a cell-specific marker). However, there are some points that remain to be clarified.1) The term "biological replicate" seems misleading because the authors seem to report that cells from only one animal were analyzed for each region. Is this standard to only use one animal for each region? It seems as though this resource would be most significant to other researchers if the reproducibility of these clusters across animals was demonstrated.

The reviewer makes an excellent comment. As a point of background, in previous work (e.g., Cembrowski et al., 2016; Cembrowski et al., 2016), we have found that animal-to-animal variability is minimal (correlation coefficient typically ~0.99 across biological replicates). This, in conjunction with the robust in situ hybridization cross-validation in our original manuscript (which uses different animals and methodologies), suggesting that our work resolved general organizational principles, rather than being animal-specific.

Nevertheless, as the reviewer notes, a demonstration the reproducibility of these clusters across animals would provide direct evidence for this general organization. To this end, we have performed sequencing of additional mice; i.e., additional biological replicates. These results include two new biological replicates for each hippocampal region, as well as additional replicates for projections; in total, our revised manuscript now includes single-cell RNA-seq data from 11 animals.

We approached this new dataset as an independent test of our original submission: employing the same experimental and computational approaches as in our original manuscript, we independently generated and analyzed this new dataset, and compared the results to our original dataset. Critically, the clusters identified in our initial submission give a near-perfect registration to the clusters in our newly acquired dataset (see new “Replicate cross-validation” subsection and Figure 2—figure supplement 3). This consistency across biological replicates, in conjunction with the histological validation presented in our original manuscript and augmented in our revision, provides strong evidence that we have uncovered robust subclasses of subiculum neurons.

2) In the Discussion, the authors state that they were "agnostic to the functional correlates" of the genes that serve to differentiate cell clusters. However, this should perhaps be explained earlier in the text, specifically in the section of the Results that describes Figure 2. Otherwise, readers may naturally wonder whether these gene expression patterns provide insights about functions of different cell groups.

Done. We now discuss functional correlates in the Results section associated with Figure 2.

Reviewer #2:[…] 1) The text should be clarified as to whether there are 8 or 9 clusters of cell types.

We thank the reviewer for emphasizing this clarification. Our scRNA-seq dataset identifies 9 clusters, one of which we ultimately determine corresponds to CA1 pyramidal cells. As such, we identity 8 clusters of subiculum neurons. We have rephrased the subsection “Spatial deconstruction of the subiculum” to explicitly clarify this point.

2) Can any of the clusters be validated with immunostaining or transgenic mouse lines?

We thank the reviewer for this suggestion, which we have now addressed.

For immunostaining, we have identified seven protein products that are enriched in subclasses of subiculum pyramidal cells, correctly predicted from our scRNA-seq data. The functional correlates of these protein products span many neuronally critical functions, including intrinsic electrophysiological properties (e.g., Kv4.3 channels), synaptic transmission (e.g., Vglut2 and synaptotagmin-2), and calcium handling (e.g., S100 and Pcp4). This provides strong evidence that our scRNA-seq accurately predicts differential protein products, which in turn likely underpin functional differences in subiculum subclasses. This work now constitutes the new Figure 8 in our revised manuscript.

For transgenic mouse lines, we have identified that our scRNA-seq work can be used to accurately identify mouse lines that yield subclass-specific access. Specifically, our scRNA-seq identified subclass-specific expression of *Slc17a6.* We show that this differential expression leads to Cre expression in *Slc17a6*-expressing subclass, and that this can be used to selectively access this population with Cre-dependent viruses. This work now constitutes our new Figure 8—figure supplement 1 in our revised manuscript.

3) In the final Results section describing projection specific-correlates of transcriptomic clusters, it is difficult to evaluate the strength of the data. How many neurons were labeled/profiled in each projection class?

We now provide these numbers in the Materials and methods for both our initial dataset as well as our new replicate dataset.

How complete was the coverage of the target area?

For each downstream target, we selected one injection site located along the anterior-posterior axis. This strategy was specifically chosen to avoid potential off-target labeling associated with injecting large volumes of the brain (which may result in spillover into spatially adjacent downstream targets, or in the case of the amygdala, into the subiculum itself). This naturally reduces the coverage of the downstream region (we estimate we cover ~10-20% of the downstream region using this strategy), but avoids confounds of off-target effects. We now note this in the Materials and methods.

The Materials and methods section indicates retro beads and AAV-SL1-CAG-tdTomato were used to label projections but in the Results section, it seems like just beads were used. This should be clarified.

Retrobeads were used for the targeted harvesting of cells for scRNA-seq, as our previous work has shown that retrobeads do not affect gene expression (Cembrowski et al., 2016). For post hocISHvalidation, AAV-SL1-CAG-tdTomato was used, as retrobead labeling is lost during ISH processing. This is now noted in the Materials and methods.

A visual for how the projection neurons map onto the spatial domains described in Figure 6 and layers in Figure 4, would be useful.

Done. This now comprises Figure 9—figure supplement 3.

Reviewer #3:[…] The potential general interest of the study is that single-cell RNAseq is relatively new and has yet to be applied to the subiculum, which is an interesting brain area because of its roles in memory. The technique is powerful in that it can identify neurons based on transcriptional signatures and when combined with projection and positional identities it can provide a convincing picture of cell diversity in a brain area. The limitations of the present study are that the claimed conceptual advances are over-stated and that the utility of the resource is compromised by a lack of replicates and limited validation.

As discussed at the very beginning of our response, we have increased our biological replicates for scRNA-seq, and also have greatly expanded our cross-validation with additional in situhybridization, immunohistochemistry, transgenic mouse lines, and comparisons to previous literature. We hope that this helps to ease the reviewer’s concerns about limited validation.

We respectfully disagree that our conceptual advances are overstated. We state our case for this below (see comment 1), and hope that by better framing of our results here and in the revised manuscript, our conceptual advance is more apparent relative to previous work.

1) While the marker genes identified will be useful, layer topography and distinctions in the proximo-distal axis of the subiculum were already largely known. For example, Ishihara and Fukuda (Neuroscience, 2016) have provided strong evidence for a layered molecular organization of the subiculum (nicely summarised in Figure 15 of their paper) and already provide good evidence for the main conclusion expressed in the title of the present study. While their study is very briefly mentioned, its convincing, important and original contributions are not sufficiently acknowledged. They should be outlined in the Introduction and the present data should be compared more directly with their results in the Discussion.

This is an excellent point. We agree with the reviewer’s point that Ishihara and Fukuda indeed make an important and original contribution that we did not sufficiently acknowledge. As such, we have incorporated additional references and comparisons to this work in our Introduction, Results, and Discussion.

However, it is important to note the scope of the Ishihara and Fukuda paper, which is based upon IHC detection of a select few proteins. Examining a small number of markers can be *suggestive* of an underlying organization – of which Ishihara and Fukuda very elegantly provide evidence – but cannot definitely resolve the underlying organization. Indeed, we have previously shown that bias introduced by examining only a few markers can fundamentally misidentify the underlying organizational scheme (Cembrowski et al., 2016). Thus, we would argue that “good evidence for the main conclusion” is an overreach of the previous work. In our revised manuscript, we have sought to reinforce the insight of Ishihara and Fukuda’s paper while also emphasizing how our work provides much stronger and comprehensive evidence for a discrete subclass organization.

It is particularly important to clarify whether the protein markers they identify (e.g. NOS, PCP4, ZnT3) are useful markers at the mRNA level based on the present data.

We now include this analysis in Figure 3—figure supplement 4. As expected from protein-level work, *Nos, Pcp4*, and *Slc30a3* (encoding ZnT3) all correspond to specific laminae in the proximal subiculum.

It may also be interesting to address whether the subiculum is analogous to the deep layers of the isocortex or neocortex.

We agree that this is a very compelling line of inquiry; however, we feel that a formal discussion is premature at this point. Our paper is primarily focused on the transcriptomic organization of the subiculum; the corresponding organization in the neocortex – especially across neocortical regions – is still very much under-resolved. As such, it is difficult to make a comparison while missing crucial features of one the comparative elements.

2) The major potential contribution of the present study will likely be for groups wanting to investigate how gene expression patterns in each of the identified subclasses relate to neuronal function or signaling cascades. For this, we find a biological replicate of 1 is not sufficient. There is a real concern here that the results are not definitive, and the study may mislead rather than benefit future work. Since the authors invested effort in making the dataset public, biological replicates would enhance the reliability of their study once it goes in the public domain.

The reviewer’s point is well-taken. As discussed in the very beginning of our response, as well as in our response to reviewer 1’s first comment, we have expanded the biological replicates for all scRNA-seq datasets. The results of this dataset recapitulate our initial findings, and are part of our new Figure 2—figure supplement 3 and Figure 9—figure supplement 1.

3) There seems to be an overlap in expression of the marker genes expressed by neurons projecting to each target area (Figure 7). For example, neurons projecting to the PFC express nearly all genes enhanced in all clusters, which weakens the conclusion that distinct transcriptional identities map onto projection choice. Similarly, NA and amygdala projection neurons express both Dlk1 and Gpc3, albeit at different proportions.

We certainly agree that there is not a perfect one-to-one relationship between projection classes and transcriptomic subclass in general. However, we do show that such a relationship can indeed exist for specific projections (e.g., the amygdala). In our revised manuscript, we have modified our wording to better reflect the different types of transcriptomic organizational principles that can apply to projection classes.

Perhaps with more animals sampled, the level of expression of marker genes will appear more even or more different? We suggest to either increase the number of animals in the sequencing assay, or to perform immunostaining / in situs to find out whether neurons projecting to each area are positive for subclass markers.

As requested, we have doubled the number of animals (i.e., biological replicates) used for sequencing each of the projection subclasses. These new replicates recapitulate the general organizational principles of our original dataset (as described in the previous paragraph) and now constitute our new Figure 9—figure supplement 1.

4) Selected genes in 5 out of the 8 clusters identified are lamina-specific, but some genes even though they belong to separate clusters seem to be expressed in an overlapping manner. For example, Cbln4 looks like it is in the same lamina as Dlk1 from single in-situs (Figure 4B). If the authors want to show this gene really marks a distinct class, they need to perform fluorescent ISH for this gene and Dlk1.

Done. As predicted by our scRNA-seq, using ISH we now show directly that these are adjacent but non-overlapping subclasses (new Figure 5D).

With this ISH successfully performed, in the spirit of having a complete examination of all ventral marker genes, we also performed a new ISH experiment examining *Ly6g6e* and *Tpbg* expression (new Figure 5E). As expected from our scRNA-seq work, these genes are associated with different laminae. We thank the reviewer for motivating us to perform this comprehensive examination of marker genes.

For the same reason, Col5a and Dlk1, which are depicted as non-overlapping in Figure 6, need validation with double ISH.

The adjacency between *Col5a2* and *Dlk1*, as illustrated from both ISH (Figure 4) and the associated schematic (current Figure 7; previously Figure 6), is minimal. This is best exemplified when comparing *Col5a2* and *Dlk1* side-by-side; note that where both genes are expressed within the same section, they tend to respectively occupy dorsal and ventral poles.

Critically, the dorsal vs. ventral enrichments alone provide a strong illustration that *Col5a2* and *Dlk1* are expressed in different populations.

To illustrate this within the same tissue sections, we next performed two-color fISH. As would be expected from minimal adjacency (Author response image 1), we found it challenging to find a local region that contained robust expression of both genes. We show an example from the intermediate subiculum (Author response image 2). Importantly, even when quantifying intermediate regions (i.e., ignoring poles in which expression is completely dominated by one gene, Author response image 1), we found that 84% of cells (n = 212 cells total from 2 animals) exhibited mutually exclusive expression of either *Dlk1* or *Col5a2* (note that *Col5a2* expression is primarily nuclear). This reinforces that *Dlk1* and *Col5a2* are markers for distinct populations of cells.

**Author response image 1. respfig1:** Side-by-side comparison of *Col5a2* and *Dlk1* across the subiculum. ISH images from Figure 3 are shown.

**Author response image 2. respfig2:** Two-color fISH of *Dlk1* and *Col5a2*. Representative image of expression of *Dlk1* (green) and *Col5a2* (magenta) in an intermediate region of the subiculum.

Also, Tpbg is depicted as restricted to the ventral half of the subiculum, even though it appears to be expressed more dorsally (Figure 3).

We thank the reviewer for pointing this out, and have now revised this figure.

A diagram indicating which genes are expressed in which lamina may also be helpful – this is hard to see in Figure 6.

We have slightly modified the coloring convention here to help interpret of this figure.

5) In the Abstract, the claim "the subiculum pyramidal cell population can be deconstructed into eight discretely separable subclasses" is difficult to justify based on the data. There is no test of discreteness in the analysis and the t-SNE plots suggest that the suggested subclasses overlap / are contiguous.

We indeed have a test for the discrete separation of clusters, examining the robustness of clusters upon downsampling (Figure 2—figure supplement 1B). This analysis illustrates that clusters are well separated, as using ~1/3 of our dataset was sufficient for ~80% success in predicting cluster identity. Such algorithms generally fail when clusters are not well-separated; the proper classification of our dataset despite a 70% downsampling is a strong indicator of effective separation.

Regarding the statement “t-SNE plots suggest that the suggested subclasses overlap / are contiguous”, this is a misinterpretation of t-SNE. t-SNE uses nonlinear dimensionality reduction to effectively balance local and global distances, and thus the distance between clusters cannot be taken as a measure of discreteness. Moreover, t-SNE is solely a visualization procedure, with clustering performed independently of this visualization. The robustness-to-downsampling analysis we discuss above is a stronger measure of separation.

Finally, the gold standard of putative discrete separation is cross-validation by in situhybridization to show nonoverlapping populations. We had multiple examples of this in our original manuscript, and have since augmented this with additional validation (see Figure 5, including the specific pairings sought in comment 3 above).

6) Choices made in setting up the analysis pipeline, and the consequences of different choices, are not sufficiently explained.

In general, this point reflects one of the broad challenges of “Big Data” in the transcriptomics community – how does one choose parameters inherent to analysis? To date, there is no well principled “one-size-fits-all” approach to choosing parameters. Our approach – philosophically and practically – has always been to analyze our data to the depth that it can be successfully validated by other techniques. As we hope the reviewer will appreciate, in our revised manuscript we have cross-validated our results extensively with single- and multicolor fluorescence in situhybridization as well as via immunohistochemistry. We also now provide a more formal phrasing of our general approach described above.

For example, to what extent does changing the cutoff thresholds in FindVariableGenes affect the subsequent clustering?

The clusters that we obtained were robust to these changes. For example, three-fold changes up or down in any the thresholds associated with this analysis (*x.low.cutoff*: minimal average expression, *x.high.cutoff*: maximal average expression, *y.cutoff:* minimum dispersion) return largely identical clusters, as shown in Author response image 3.

**Author response image 3. respfig3:** Robustness of clusters across threshold values. Each row corresponds to changing the value of a given parameter inherent to the FindVariableGenes call; specifically, *x.low.cutoff* (top row), *x.high.cutoff* (middle row), and *y.cutoff* (bottom row). Values explored are 3-fold decrements (left column) and 3-fold increments (right column) relative to manuscript value (middle column). Left to right, this constitutes nearly an order of magnitude difference in the parameter value. T-SNE visualizations and clustering results are shown for each parameter regime. Colours denote clusters obtained by graph-based approach used in main manuscript, with colouring associated with marker genes in main manuscript. Note, although in principal such lusters may be strongly affected by changes in these cutoff values, in practice they are not: marked stability is present in the number of clusters, the number of cells per cluster, and the marker genes associated with each cluster.

We now explicitly mention this robustness formally in our Materials and methods.

We hasten to note that, in addition to the above robustness (a measure of *internal* consistency of our original RNA-seq dataset), in our revised manuscript we also have multiple measures of the more stringent test of *external*consistency with other datasets. Specifically:

1) The clusters obtained in our original dataset map onto clusters generated in a new biological replicate dataset (new Figure 2—figure supplement 3);

2) The clusters obtained in our original dataset map onto discrete subclasses confirmed by in situhybridization (newly expanded Figure 5);

3) The clusters obtained in our original dataset map onto predicted protein products (new Figure 8).

To what extent is clustering affected by the resolution parameter?

Changing the resolution parameter naturally controls the degree of “lumping” vs. “splitting”. The value we used in our manuscript produces a set of clusters that are consistent with multiple cross-validations. Further splitting leads to clusters that harder to cross-validate, which may emerge due to oversplitting or may reflect more nuanced biology. We have erred on the side of being conservative in our calls, requiring that all clusters be supported by both RNA-seq data and in situhybridization. We now explicitly mention this formally in our Materials and methods.

In generating the t-SNE plots what were the perplexity values and how many steps were used?

We used a standard value of perplexity (=30), and 1000 iterations which were sufficient for convergence. We now list these in the Materials and methods.

Several sub-classes do not appear particularly well separated in the t-SNE plots. It may be that optimising perplexity or steps gives a better indication of separation, or it may be that they are not separable even with t-SNE.

As we noted above (point 5), the idea that clusters need to be well-separated in t-SNE space for discrete separation is a misinterpretation of t-SNE. Changing parameters in t-SNE, which is solely a visualization procedure and has nothing to do with formal clustering per se, will not change any clustering results. We have both analytical controls (Figure 2—figure supplement 1B) and extensive histological cross-validation (newly expanded Figure 5) that directly illustrate clusters that appear “close” in t-SNE space are indeed discretely separated.

7) Figure 4C-E and Figure 7—figure supplement 1 require quantification of the numbers of cells and numbers of biological replicates.

In our previous version of the manuscript, these quantifications were provided in-line within the Results section (number of cells) as well as in the associated Materials and methods (number of biological replicates). In our revised version, for ease of readability we have combined these details in the Results section for Figure 5 (previously Figure 4) or the legend for Figure 9—figure supplement 2 (previously Figure 7—figure supplement 1).

8) Figure 2—figure supplement 1B suggests that approximately 20-25% of cells are not classifiable. Does this population overlap across simulation runs?

We have performed this analysis and include the results as Author response image 4. As would be expected, most cells that are misclassified typically occupy the extremes of clusters, and do exhibit overlap across runs. Smaller clusters (e.g., cluster 9) also tended to exhibit higher rates of misclassification, owing to smaller representation in the overall dataset (note that minority class prediction is a general challenge for machine-learning algorithms).

**Author response image 4. respfig4:** Response Figure 4. Cellular resolution of cluster assignment. A total of 1000 stochastic simulations were run, wherein for each simulation, 800 cells were stochastically selected for training a random forest classifier and the remaining 303 cells in the dataset were used for testing this classifier (as in Figure 2—figure supplement 1B). Each cell is colored according to the percent of success identifications across test trials. For comparison, cluster designation is provided at right. Most cells exhibited near-perfect assignment for the correct clusters, with misassigned cells typically occupying the extrema of clusters. Notably, cluster 9 typically exhibited poor assignment, likely arising from its small and underrepresented nature in the dataset (24 cells or ∼2% of dataset; conventional random forest classifiers are biased against underrepresented classes). B. For comparison, the tSNE visualization of clusters is provided.

If so, what happens to the cell classification if these cells are excluded? For example, are the groupings cleaner?

Certainly, elimination of these cells would help to separate groupings, as by definition one is removing points that have ambiguous classification. However, such a post hoc elimination of undesired data points is not a statistically well-principled way to approach analysis. We would prefer not to mask any cells from our analysis in this way, and instead have expanded upon misclassification in Figure 2—figure supplement 1.

9) For reproducibility, the main analysis code should be made available alongside the study.

We have an established track-record of providing all code associated with each of our published RNA-seq studies. All code used in this study will be available upon acceptance, as stated in both our original and revised manuscripts.

When doing so it would be helpful to make explicit the random number seeds used by Seurat and the version of Seurat used. When we used Seurat to analyze the dataset we obtained different, although qualitatively similar, plots to those in the manuscript. We guess this is because of differences in seeding but cannot know for sure. The methods should also make clear whether key analyses were repeated with different seeds and whether similar results were obtained.

Where the reviewer refers to “seed” here, we assume that this is a reference to t-SNE, which is the only element of our analysis pipeline that contains stochasticity and is shaped by seeding. We used the default seed for our t-SNE visualization, and have run this visualization with multiple seeds and have obtained qualitatively similar results. This is now stated explicitly in the Materials and methods section.

[Editors' note: further revisions were requested prior to acceptance, as described below.]

Essential revisions:1) The revisions do more to acknowledge previous work by Ishihara and Fukuda. However, this previous work is still not credited sufficiently. This doesn't affect the validity of the conclusions, nor the strong case for publishing the study, but a more scholarly approach to recognizing previous work would reflect well on the authors and the journal. We have some suggestions.Introduction, third paragraph, acknowledges Ishihara and Fukuda, but their contribution is not made sufficiently clear here. Perhaps state, "Recent investigations using immunohistochemical labeling argue that the proximal subiculum is composed of a molecular layer and multiple cell body layers, each distinguished by molecular and morphological differences, while the distal subiculum is more uniform (Ishihara and Fukuda, 2016)."Discussion, subsection “The subiculum as a laminar and columnar structure”, last paragraph. It's important to be clear here that prior evidence for and articulation of the idea that "the subiculum can be deconstructed into proximal and distal subdomains, with further laminar organization predominantly found in proximal subiculum" was provided in the previous study by Ishihara and Fukuda. We agree that the present results further refine their model, but as written the text could easily be interpreted as taking credit for ideas previously proposed by Ishihara and Fukuda.

We now explicitly note that this was previously proposed by Ishihara and Fukuda.

Also, the implication in the first paragraph of the aforementioned subsection, that Ishihara and Fukuda's scheme was based only on protein labeling is incorrect, as they also took account of neuronal morphology, something the present study doesn't do.

We now note that Ishihara and Fukuda analyzed neuronal morphology.

2) We're not convinced by the conclusion that the analysis supports the existence of eight discrete cell classes.

Although all computational and experimental results we have obtained are consistent with the existence and discrete separation of these 8 classes, it is possible in principle that some continua may exist between specific subclasses. As the following 3 points raised by the reviewer below ultimately culminate in “These problems could easily be addressed by [several means of improvement]”, we keep the rest of our response to the end of this section.

It seems pretty clear that some of the cell classes are distinct (e.g. groups labeled by Cbln4 vs. Dlk1), but while others can be grouped, the possibility that rather than being discrete classes they reflect a split along a continuum, or overlapping classes, is not convincingly addressed (e.g. S100b vs. Tpbg).First, it's not clear how the algorithm used for clustering provides evidence for groups being generated by discrete as opposed to continuous distributions. As the authors recognize this is a difficult problem. Clustering algorithms will group some continuous distributions into clusters, but this is not evidence that the distributions were in fact discrete. Rather than under-sampling, which doesn't address the problem convincingly, a more rigorous statistical approach is required to support the conclusion that there are eight discrete groups. Previous comments about the interpretation of the tSNE plots must not have been sufficiently clear. They of course don't test for discreteness. Rather, a typical concern is that tSNE can give an exaggerated sense of distances between data points, whereas here even when visualizing the data using tSNE plots the groups appear contiguous.Second, the new data in Figure 5 are nice, but their interpretation requires clarification. Each comparison is between groups that appear well separated by the clustering algorithm (cf. Figure 2A) and the conclusion that the groups labeled in Figure 5 are distinct seems very reasonable. However, the clusters in Figure 2A that appear contiguous (e.g. S100b and Tpbg, Dlk1 and Lefty1, Dlk1 and Gpc3, etc.) are not validated in Figure 5. Thus, Figure 5 doesn't support the idea that all eight proposed cell groups are discretely separable, as there is no evidence here that the adjacent groups of cells from Figure 2A are separable.Third, the argument that ~80% successful classification with a random forest validates the groupings is not by itself convincing. For genuinely discrete classes wouldn't you expect the RF to do a little better? Are the misclassified cells randomly distributed, or are errors more systematic? For example, are all groups misclassified at the same rate, or is the error rate higher for particular groups? Are some types of error over-represented, e.g. does S100b misclassify as Tpbg, and vice -versa, at higher rates than other pairings? What error rates would you expect given alternative hypotheses that some groups, e.g. S100b and Tpbg, represent a split along a continuum, rather than discrete classes?These problems could easily be addressed by avoiding use of "discrete" as a conclusion, e.g. in the title, in the Abstract, and multiple other parts of the manuscript.

In our revised manuscript, we have now removed the word “discrete” as it refers to the scRNA-seq results, and use this word to reference our work only in cases where we have explicitly confirmed discreteness via cross-validation (e.g., in situhybridization). We have kept this word in the title of our work because it is an accurate representation of our body of work as a whole – even if some scRNA-seq subclasses do actually exist in a continuum, they will nonetheless be discretely separated from other subclasses (as well as be in the minority of all subclasses; see new Discussion paragraph).

By discussing the potential ambiguities, and by being open to the possibility that some, although clearly not all, of the groupings may not be discrete. Along the same lines, the idea that there are eight groups would be better phrased as a suggestion rather than a definitive conclusion.

In our revised Discussion we have now included an additional paragraph on the interpretation of our scRNA-seq-derived subclasses. This paragraph covers the possibility of discrete vs. continuous subtypes, as well as the total number of subtypes of subiculum pyramidal cells.